- 1 Living on the edge: Response of Late Cretaceous rudist bivalves (Hippuritida) to hot and highly seasonal
- 2 climate in the low-latitude Saiwan site, Oman
- Niels J. de Winter<sup>1,2</sup>, Najat al Fudhaili<sup>3</sup>, Iris Arndt<sup>4</sup>, Philippe Claeys<sup>2,5</sup>, René Fraaije<sup>6</sup>, Steven Goderis<sup>2</sup>, John
- 4 Jagt<sup>7</sup>, Matthias López Correa<sup>8</sup>, Axel Munnecke<sup>8</sup>, Jarosław Stolarski<sup>9</sup>, Martin Ziegler<sup>10</sup>

6

7

10

11

14

15

## Affiliations

- 1. Department of Earth Sciences, Vrije Universiteit Amsterdam, Amsterdam, the Netherlands
- Archaeology, Environmental Changes and Geo-Chemistry, Vrije Universiteit Brussel, Brussels,
   Belgium
  - 3. Industrial Innovation Academy LLC, Muscat, Oman
  - 4. Institut für Geowissenschaften, Goethe-Universität Frankfurt, Frankfurt, Germany
- 5. Pacific Centre for Isotopic and Geochemical Research, Department of Earth, Ocean and Atmospheric Sciences, University of British Columbia, Vancouver, BC, Canada
  - 6. Het Nationaal Oertijdmuseum Boxtel, Boxtel, the Netherlands
  - 7. Natuurhistorisch Museum Maastricht, Maastricht, the Netherlands
- 16 8. GeoZentrum Nordbayern, Friedrich-Alexander Universität, Erlangen, Germany
- 9. Institute of Paleobiology, Polish Academy of Science, Warsaw, Poland
- 10. Department of Earth Sciences, Utrecht University, Utrecht, the Netherlands

Corresponding author: Niels J. de Winter (n.j.de.winter@vu.nl)

22

## Abstract

- Earth's climate history serves as a natural laboratory for testing the effect of warm climates on the 24 biosphere. The Cretaceous period featured a prolonged greenhouse climate characterized by higher-than-25 modern atmospheric CO₂ concentrations and mostly ice-free poles. In such a climate, shallow seas in low 26 latitudes probably became very hot, especially during the summers. At the same time, life seems to have 27 thrived there in reef-like ecosystems built by rudists, an extinct group of bivalve molluscs. To test the 28 seasonal temperature variability in this greenhouse period, and whether temperature extremes exceed 29 the maximum tolerable temperatures of modern marine molluscs, we discuss a detailed 30 sclerochronological (incrementally sampled) dataset of seasonal scale variability in shell chemistry from 31 fossil rudist (Torreites sanchezi and Vaccinites vesiculosus) and oyster (Oscillopha figari) shells from the 32 late Campanian (75-million-year-old) low latitude (3°S paleolatitude) Saiwan site in present-day Oman. We 33 combine trace element data and microscopy to screen fossil shells for diagenesis, before sampling well-34 preserved sections of a Torreites sanchezi rudist specimen for clumped isotope analysis. Based on this 35 specimen alone, we identify a strong seasonal variability in temperature of 19.2 ± 3.8°C to 44.2 ± 4.0°C in the seawater at the Saiwan site. The oxygen isotopic composition of the seawater ( $\delta^{18}O_{sw}$ ) varied from -36
- We use this information in combination with age modelling to infer temperature seasonality from

 $4.62 \pm 0.86$  % VSMOW in winter to  $\pm 0.86 \pm 1.6$  % VSMOW in summer.

incrementally sampled oxygen isotope profiles sourced from the literature, sampling multiple shells and

species in the assemblage. We find that, on average, the Saiwan seawater experienced strong seasonal fluctuations in monthly temperature (18.7  $\pm$  3.8 to 42.6  $\pm$  4.0 °C seasonal range) and water isotopic composition (-4.33  $\pm$  0.86 to 0.59  $\pm$  1.03 %VSMOW). The latter would strongly bias the interpretation of stable oxygen isotopes in shell carbonate without independent control on either temperature or seawater composition.

Combining our seasonal temperature estimates with shell chronologies based on seasonal cyclicity in stable isotope records and daily variability in trace element data, we show that *T. sanchezi* rudists record temperatures during the hottest periods of the year as well as during the winters, which were characterized by cooler temperatures and enhanced influx of freshwater. Both *O. figari* and *V. vesiculosus* plausibly stopped growing during these seasonal extremes. This study aims to demonstrate how high-resolution geochemical records through fossil mollusc shells can shed light on the variability in past warm ecosystems and open the discussion about the limits of life in the shallow marine realm during greenhouse climates. Future work should apply the clumped isotope paleothermometer on incrementally sampled bio-archives to explore the upper-temperature limits experienced by calcifiers in different environments throughout geological history.

#### 1. Introduction

Ongoing anthropogenic global changes, including greenhouse gas emissions and land use changes, are projected to increase global mean annual temperatures by multiple degrees with respect to pre-industrial conditions, while at the same time causing severe biodiversity loss (IPCC, 2023; World Wildlife Fund, 2020). These crises are intricately linked, but assessing the effect of climate change on biodiversity loss requires information on the response of biodiversity to climate extremes under various (paleo)climate scenarios. The geological record provides a rich source of such information in the form of fossil bio-archives that record climate and environmental change on the scale of days to decades, while testifying to biodiversity by their presence in the rock record (Huyghe et al., 2012; Ivany, 2012; Schmitt et al., 2022). Past ecosystems thus serve as natural experiments for testing the limits of life during periods of global change or in exceptionally warm periods (Cermeño et al., 2022; de Winter et al., 2017; Jones et al., 2022).

Examples of hot periods that may reveal ecosystems' functioning under high-temperature climate scenarios include the early Triassic super greenhouse (Sun et al., 2012), the Mid-Cretaceous Climate Optimum (Jones et al., 2022) or the Eocene hothouse period (de Winter et al., 2020; Evans et al., 2013). Milder, yet still warmer than present-day, scenarios of interest include the Late Cretaceous ( de Winter et al., 2020; O'Hora et al., 2022; Petersen et al., 2016a), the Miocene Climatic Optimum (Batenburg et al., 2011; Harzhauser et al., 2011) and the Pliocene Warm Period (de Winter et al., 2024; Dowsett et al., 2013; Wichern et al., 2023). While these periods feature long-term, equilibrated climate states instead of fast, transient climate change events (like modern anthropogenic warming), they can yield useful insights into the long-term response of the climate system and biosphere to prolonged radiative forcing (Burke et al., 2018). For example, some of these past environments, most notably shallow marine ecosystems, are thought to have reached temperatures exceeding the temperature range of modern equivalent ecosystems (de Winter et al., 2020; Huang et al., 2017; Jones et al., 2022). These temperatures probably exceeded the maximum temperature tolerance at which modern shallow marine species can complete their life cycle, which is typically estimated in the order of 38-42°C (Compton et al., 2007). ) Conditions that exceed this threshold (>38°C) are considered high-temperature conditions for shallow marine calcifiers to live in.

A striking conundrum arises in the Late Cretaceous Tethys Ocean margins, which were inhabited by large rudist bivalves biostromes (Skelton, 2018) despite apparently high water temperatures. The atmospheric CO<sub>2</sub> concentrations during the Campanian (83.6 – 72.1 Ma (Gradstein et al., 2020)) were ~600 ppmV (roughly 2x pre-industrial concentrations (Wang et al., 2014)), resulting in low- to mid-latitude mean annual sea surface temperatures (SST) of 20-25°C (O'Brien et al., 2017), roughly 5-10°C warmer than the current global mean annual temperature. In low-latitude Tethyan margins, mean annual temperatures likely exceeded 30°C (Steuber et al., 2005), with summer temperatures estimated above 40°C in some localities (de Winter et al., 2017, 2020; Steuber, 1999). A big caveat of these estimates is that they rely on the temperature dependence of stable oxygen isotope analyses and are contingent on assumptions of (seasonally) constant stable oxygen isotope composition of Tethyan seawater, which is known to have fluctuated over time (Price et al., 2020; Walliser and Schöne, 2020). If correct, such temperatures exceed the temperature threshold of 38°C mentioned above and are at or above the lethal thermal limits for modern marine invertebrates (typically 42-50°C; (Clarke, 2014; Compton et al., 2007)). These temperatures approach the thermal threshold above which critical macromolecules such as ATP, proteins, and enzymes used by non-extremophile organisms denature (>50°C) (Clarke, 2014; Tehei et al., 2005; Tehei and Zaccai, 2007), hampering key metabolic functions. Yet, despite these apparent temperature extremes,

abundant and diverse fossil shallow marine ecosystems suggest that rudists thrived in these environments (Gili and Götz, 2018; Ross and Skelton, 1993). This raises the question of whether these paleotemperature reconstructions are accurate, and if so, whether these ancient molluscs were somehow adapted to grow their shells at these extreme temperatures.

In an attempt to resolve this thermal tolerance conundrum, we investigate the growth and chemical composition of two species of rudist bivalve (Torreites sanchezi and Vaccinites vesiculosis) and one oyster species (Oscillopha figari) from the low-latitude Tethyan site in the Saiwan area in east-central Oman (Kennedy et al., 2000; Schumann, 1995). Our analysis combines new sclerochemical data, including clumped isotope analyses, with existing stable oxygen isotope datasets from the literature. We aim to test how the growth of these animals responds to seasonal temperature extremes. To obtain accurate seasonal temperature reconstructions that are independent of the Tethyan seawater composition and ex vivo diagenetic alteration, we combine information on trace elements, stable oxygen isotopes, and clumped isotopes. Using a new clumped isotope dataset, we first demonstrate that T. sanchezi records temperatures in their shells that significantly exceed the threshold at which modern marine molluscs thrive. We then build on our clumped isotope dataset to test how shell growth was influenced by these seasonal temperature extremes in this paleoenvironment by combining our results with stable oxygen isotope records from the literature, augmented with new shell chronologies and growth models. We contrast our seasonality and growth rate reconstructions with data on the thermal ranges of modern bivalves to compare the tolerance of biostrome-building rudists in the Late Cretaceous with modern species. Ultimately, this case study will open the discussion of the ability of marine life to adapt to hot climates and provide lessons for the interaction between climate change and marine biodiversity.

### 2. Materials and Methods

# 120 2.1 Fossil mollusc specimens

The mollusc shells utilized in this work originate from the Samhan Formation in the Saiwan area of Oman 122 in the Huqf desert (30°39 N, 57°31 E; see Figure 1A). The biostromes in this locality were dated as late 123 Campanian (~75 Ma) based on ammonite biostratigraphy by Kennedy et al. (2000). The paleolatitude of 124 the site at 75 Ma was 3°S according to reconstructions following Paleolatitude.org (van Hinsbergen et al., 125 2015) based on the paleomagnetic reference frame by Vaes et al. (2023). The locality was described by 126 (Schumann, 1995) as exposing Vaccinites-dominated rudist biostromes in which rudist bivalves are 127 preserved in their life position (see also Figure 1B-C). The V. vesiculosus and T. sanchezi specimens used 128 and reused in this study originate from Unit IV in profile 1 of Schumann (1995), which is equivalent to unit 129 2 in (Philip and Platel, 1995). The thick-shelled O. figari oysters were collected in the marly layer just above 130 the biostrome (see also Figure 1B-C and de Winter et al. (2017)).

Figure 1: Showing (A) Map of the geographical location of the Late Campanian Samhan Formation in the Saiwan area in the central part of the Sultanate of Oman. (B) Lithostratigraphic column showing the stratigraphic context of the characteristic members of the Samhan Formation. The red dashed box indicates the stratigraphic location of the outcrop pictured in (C). The marly layer containing *O. figari* referred to in the text is the brown layer in between the two green members in the dashed red box. (C) Outcrop showing the two *Vaccinites*-dominated biostromes, the lower of which contained the *T. sanchezi* and *V. vesiculosus* specimens investigated in this study (indicated by the schematic images of the species). The *O. figari* specimen was collected in the marly layer just above the biostrome containing the other specimens (indicated by the schematic image), between the two green units in (B). Panel (B) was modified after (Philip and Platel, 1995).

All specimens described in this study were sampled in longitudinal cross-sections through the axis of maximum growth using a combination of hand drilling and computer-assisted microdrilling using slow-

rotating tungsten carbide dental drills. **Table 1** gives an overview of the data used in this study, its temporal resolution and the source of datasets in cases where they have been reused from the literature. *V. vesiculosus* specimen "**B6**", *T. sanchezi* specimen "**B10**" and *O. figari* specimen "**B11**" were described in ( de Winter et al., 2017) and stable isotope ( $\delta^{18}$ O and  $\delta^{13}$ C) and trace element (Mg, Sr, Ca, Mn, Fe) data presented in that study are used here. Specimen **B10** was also subject of a study by (de Winter et al., 2020), in which more detailed, daily scale trace element measurements (Mg/Ca, Sr/Ca, Mg/Li and Sr/Li) were presented and discussed. Stable isotope ( $\delta^{18}$ O and  $\delta^{13}$ C) data from *T. sanchezi* specimens "**H576**", "**H579**" and "**H585**" was previously reported in (Steuber, 1999). Of these, specimen **H579** was sampled in 5 parallel sections ("**H579A-E**").

For this study, we obtained newly measured stable ( $\delta^{18}O$  and  $\delta^{13}C$ ) and clumped ( $\Delta_{47}$ ) isotope analyses on one additional *T. sanchezi* specimen, called "HU-027". This specimen was incrementally sampled (n = 135) at 50 µm resolution for  $\delta^{18}O$  and  $\delta^{13}C$  in a cross-section in the internal pillar of the rudist shell. A total of 96 clumped isotope measurements were carried out on incremental samples from two locations corresponding to the maximum and minimum  $\delta^{18}O$  and  $\delta^{13}C$  values in the above-mentioned profile (samples R\_1 - R\_11) as well as larger bulk samples in two areas on the same cross-section (samples RB\_1 and RB\_2; Figure 2A). Sample RB\_2 was obtained from a different pillar in the same cross-section, whose timing of growth can be linked to the adjacent pillar by following the isochronous growth lines visible in the cross-section (Figure 2B). The timing of deposition of the shell calcite sampled in RB\_2 partly overlaps with incremental samples R\_1 - R\_4, but this sample also contains a significant portion of shell material deposited earlier in the ontogeny.

**Table 1**: Overview of data used in this study and their sources

| Specimen     | Species     | Proxy             | # data points | Resolution               | Source                         |  |  |  |  |  |  |  |
|--------------|-------------|-------------------|---------------|--------------------------|--------------------------------|--|--|--|--|--|--|--|
|              |             | $\delta^{18}O$    | 306           | ~2 weeks                 | (de Winter et al., 2017)       |  |  |  |  |  |  |  |
|              |             | $\delta^{13}C$    | 306           | ~2 weeks                 | (de Winter et al., 2017)       |  |  |  |  |  |  |  |
| D.C          | V.          | Mg/Ca             | 735           | ~1 week                  | (de Winter et al., 2017)       |  |  |  |  |  |  |  |
| B6           | vesiculosus | Sr/Ca             | 735           | ~1 week                  | (de Winter et al., 2017)       |  |  |  |  |  |  |  |
|              |             | Mn                | 735           | ~1 week                  | (de Winter et al., 2017)       |  |  |  |  |  |  |  |
|              |             | Fe                | 735           | ~1 week                  | (de Winter et al., 2017)       |  |  |  |  |  |  |  |
|              |             | δ <sup>18</sup> Ο | 310           | ~2 weeks                 | (de Winter et al., 2017, 2020) |  |  |  |  |  |  |  |
|              |             | δ <sup>13</sup> C | 310           | ~2 weeks                 | (de Winter et al., 2017, 2020) |  |  |  |  |  |  |  |
|              |             | Mg/Ca             | 12443         | ~0.5h                    | (de Winter et al., 2020)       |  |  |  |  |  |  |  |
|              |             | Sr/Ca             | 12535         | (de Winter et al., 2020) |                                |  |  |  |  |  |  |  |
| D40          | _ , .       | Mg/Li             | 12167         | (de Winter et al., 2020) |                                |  |  |  |  |  |  |  |
| B10          | T. sanchezi | Sr/Li             | 12322         | ~0.5h                    | (de Winter et al., 2020)       |  |  |  |  |  |  |  |
|              |             | Mg/Ca             | 4353          | 1-5h                     | (de Winter et al., 2017)       |  |  |  |  |  |  |  |
|              |             | Sr/Ca             | 4361          | 1-5h                     | (de Winter et al., 2017)       |  |  |  |  |  |  |  |
|              |             | Mn                | 4043          | 1-5h                     | (de Winter et al., 2017)       |  |  |  |  |  |  |  |
|              |             | Fe                | 3972          | 1-5h                     | (de Winter et al., 2017)       |  |  |  |  |  |  |  |
| H576         | T           | δ18Ο              | 98            | ~3 weeks                 | (Steuber, 1999)                |  |  |  |  |  |  |  |
|              | T. sanchezi | δ <sup>13</sup> C | 98            | ~3 weeks                 | (Steuber, 1999)                |  |  |  |  |  |  |  |
| H579         |             |                   | 288           |                          |                                |  |  |  |  |  |  |  |
|              | T. sanchezi | $\delta^{18}O$    | (116 + 46 +   | ~3 weeks                 | (Steuber, 1999)                |  |  |  |  |  |  |  |
|              |             |                   | 47 + 35 + 44) |                          |                                |  |  |  |  |  |  |  |
| (5 profiles: |             |                   | 288           |                          | (Steuber, 1999)                |  |  |  |  |  |  |  |
| A-E)         |             | $\delta^{13}C$    | (116 + 46 +   | ~3 weeks                 |                                |  |  |  |  |  |  |  |
|              |             |                   | 47 + 35 + 44) |                          |                                |  |  |  |  |  |  |  |
| LIEGE        | T           | δ <sup>18</sup> Ο | 132           | ~1 month                 | (Steuber, 1999)                |  |  |  |  |  |  |  |
| H585         | T. sanchezi | $\delta^{13}C$    | 132           | ~1 month                 | (Steuber, 1999)                |  |  |  |  |  |  |  |
|              |             | δ <sup>18</sup> Ο | 224           | 1.2 days                 | This study (BSc thesis N. al-  |  |  |  |  |  |  |  |
|              |             | 8-50              | 231           | 1-2 days                 | Fudhaili)                      |  |  |  |  |  |  |  |
| 027          | T sanahazi  | δ <sup>13</sup> C | 231           | 1.2 days                 | This study (BSc thesis N. al-  |  |  |  |  |  |  |  |
| HU-027       | T. sanchezi | 0 1               | 231           | 1-2 days                 | Fudhaili)                      |  |  |  |  |  |  |  |
|              |             | $\Delta_{47}$     | 86            | Seasonal                 | This study                     |  |  |  |  |  |  |  |
|              |             | Si & Ca           | map           | Spatial map              | This study                     |  |  |  |  |  |  |  |
|              |             | δ18Ο              | 90            | ~3 weeks                 | (de Winter et al., 2017)       |  |  |  |  |  |  |  |
|              |             | $\delta^{13}C$    | 90            | ~3 weeks                 | (de Winter et al., 2017)       |  |  |  |  |  |  |  |
| D11          | O figari    | Mg/Ca             | 402           | ~3-7 days                | (de Winter et al., 2017)       |  |  |  |  |  |  |  |
| B11          | O. figari   | Sr/Ca             | 402           | ~3-7 days                | (de Winter et al., 2017)       |  |  |  |  |  |  |  |
|              |             | Mn                | 402           | ~3-7 days                | (de Winter et al., 2017)       |  |  |  |  |  |  |  |
|              |             | Fe                | 367           | ~3-7 days                | (de Winter et al., 2017)       |  |  |  |  |  |  |  |

Specimens **H576**, **H579** and **H585** were subject to detailed diagenetic screening in (Steuber, 1999) and the preservation of specimens **B6**, **B10** and **B11** was tested in (de Winter et al., 2017, 2020; de Winter and Claeys, 2016). These previous studies concluded, based on a combination of scanning electron microscopy, cathodoluminescence microscopy and trace element analysis, that there was no detectable recrystallization in the areas of the shells sampled for geochemical analysis and that the low-magnesium

calcite outer shell layer of these rudists preserves the original shell composition deposited during the lifetime of the animal. For the purpose of this study, additional screening based on high-resolution trace element analyses (Mg/Ca, Sr/Ca, Mn and Fe) will be discussed for specimens **B6**, **B10** and **B11**, one specimen for each of the three studied species (see **sections 3.2** and **4.1**). The newly sampled specimen **HU-027** was subject to detailed microscopic scrutiny using a combination of reflected light, cross-polarized light, scanning electron microscopy and energy dispersive X-ray spectroscopy (EDS) and micro-Raman spectroscopy to characterize original shell texture preservation and detect diagenetic alteration (**Figure 2C-F**).

**Figure 2**: **A)** High-resolution colour scan of a cross-section through *T. sanchezi* specimen **HU-027** with the location of samples for  $\delta^{18}$ O,  $\delta^{13}$ C (yellow rectangle) and clumped isotope analysis (black text and lines). **B)** Zoomed-in insert showing fine lamination in columns through **HU-027**, the location of samples **RB\_2** and **R\_1** - **R\_5**, and examples of isochronous growth lines (dashed lines) that link the timing of growth between pillars in the cross-section. **C)** Cross-polarized light image of the edge of a shell column showing locations for characterizing diagenetic (silicified) and pristine areas in the shell. **D)** Raman spectra of pristine calcite (yellow) and silicified (red) areas in the shell of **HU-027**. **E)** Scanning Electron Microscopy fore scatter image of the area of interest highlighted in **B**, showing original calcite shell structures (right)

- and silicified areas of the shell (left). F) Energy-dispersive X-ray spectroscopy (EDS) image of the same area
- shown in **E** showing the silicon (red) and calcium (yellow) composition on a micrometre scale.
- 2.2 Chemical data
- 2.2.1 X-ray fluorescence
- To test the preservation of shells of the specimens **B6**, **B10** and **B11**, trace element concentrations were
- analysed in situ using a Bruker M4 Tornado (Bruker nano GmbH) micro-X-ray fluorescence (µXRF) scanner
- (see de Winter and Claeys (2016)). The M4 is equipped with a Rh-anode X-ray source, which was operated
- at maximum energy settings (50 kV, 600 μA) without an X-ray filter, and X-rays were focused on a 25 μm
- circular spot (calibrated for Mo-kα radiation) on the flat sample surface. Fluorescent X-rays were detected
- using two silicon drift detectors for maximum count rates.
- Cross-sections of the entire specimens were first mapped to determine the best preserved sampling
- localities based on Mn and Fe concentrations (see de Winter et al. (2017)). This μXRF mapping was carried
- out by moving the X-ray beam along the sample surface in a raster pattern while continuously collecting
- XRF spectra. Since this mapping mode allows to only collect fluorescent X-rays for ~1 millisecond for each
- 25 μm-wide pixel, XRF spectra of individual pixels cannot be quantified. Instead, maps were quantified as
- a whole by integrating the area under XRF peaks for all pixels, producing false-colour images of semi-
- quantitative trace element abundance across the entire specimen (see de Winter et al. (2017)).
- Quantitative XRF profiles were gathered in growth direction on cross-sections through the shells by
- analysing point-by-point line scans (see Vansteenberge et al., 2020). This point-by-point analysis allows the
- X-ray beam to dwell on a single spot for 60 seconds, allowing the detectors to gather enough XRF counts
- for a quantifiable XRF spectrum (de Winter et al., 2017). XRF data were quantified through a combination
- of peak deconvolution and fundamental parameter quantification in the Bruker Esprit software (Bruker
- nano GmbH), calibrated using the matrix-matched carbonate reference material BAS-CRM393 (Bureau of
- Analyzed Samples, Middlesbrough, UK). Quantified trace element concentrations were subsequently
- calibrated based on a set of 10 matrix-matched carbonate reference materials (see (Vellekoop et al., 2022)
- to obtain reproducible trace element concentrations. The XRF maps and line scans were used to
- demonstrate the preservation of the original calcium carbonate in specimens **B6**, **B10** and **B11**. Full XRF
- datasets for all analysed specimens are provided in **Supplement S1** in the accompanying Zenodo
- repository (https://doi.org/10.5281/zenodo.12567712).
- 2.2.2 Laser Ablation Inductively Coupled Plasma Mass Spectrometry
- High-resolution Laser Ablation-Inductively Coupled Plasma Mass Spectrometry (LA-ICP-MS) profiles of Li,
- 218 Mg, Sr, Ca, Mn and Fe concentrations through *T. sanchezi* specimen **B10** were reused from ( de Winter et
- al., 2020). These profiles were measured using an Analyte G2 ArF\*excimer-based laser ablation system
- (Teledyne Photon Machines, Bozeman, USA) coupled to an Agilent 7900 (for LA-ICP-MS; Agilent, Santa
- Clara, USA) quadrupole-based ICP-MS unit. The laser was focused on a round 10 µm spot on the sample
- surface and translated continuously in line scanning mode to gather sub-daily resolved data along the
- entire shell height. LA-ICP-MS results were calibrated using repeated measurements of United States
- Geological Survey (USGS) BCR-2G, USGS BHVO-2G, USGS GSD-1G, and USGS-GSE-1G, and National
- Institute of Standards and Technology SRM610 and National Institute of Standards and Technology

- SRM612 certified natural and synthetic glass reference materials. All LA-ICP-MS concentration data are
- provided in **Supplement S1** in the Zenodo repository.

in **Supplement S2** in the Zenodo repository.

- 2.2.3 Isotope Ratio Mass Spectrometry
- Stable carbon ( $\delta^{13}$ C) and oxygen ( $\delta^{18}$ O) values from specimens **B6**, **B10**, **B11**, **H576**, **H579** and **H585** were 229 reused from (de Winter et al., 2017) and (Steuber, 1999). This dataset was augmented by new sequentially 230 sampled  $\delta^{13}$ C and  $\delta^{18}$ O measurements along the central pillar through specimen **HU-027** (see **Figure 2**). 231 All  $\delta^{13}$ C and  $\delta^{18}$ O data were obtained by analysing carbonate powders sampled in cross-sections through 232 233 the specimens using an Isotope Ratio Mass Spectrometer (IRMS) coupled to a carbonate preparation 234 device. The stable isotope analyses for the incremental samples in **HU-027**, specifically, were analysed by 235 a Thermo DELTA V+ IRMS coupled to a Gasbench carbonate preparation device. During sampling, care was 236 taken to avoid areas in the shell characterized by elevated Mn and Fe concentrations (see trace element 237 results in section 3.2 and (de Winter et al., 2017)) and microscopic signs of diagenetic alteration (see Figure 238 2). Standard deviations of uncertainty on  $\delta^{13}$ C and  $\delta^{18}$ O values produced using this technique are 0.05 % and 0.10 ‰, respectively. An overview of the  $\delta^{13}$ C and  $\delta^{18}$ O records through all IRMS profiles is provided 239
- Representative samples from two parallel central pillars of specimen HU-027 were sampled for clumped 242 isotope analysis (see Figure 2). During sampling, care was taken to avoid the diagenetically altered sections 243 of the shell (Figure 2C-F). Two areas in HU-027 were drilled for bulk clumped isotope analysis ("RB 1" and 244 "RB\_2"; Figure 2A; see section 2.1), after which samples were drilled from transects in growth direction 245 along one of the pillars exposed in the shell (Figure 2A). A total of 94 small (70-95 µg) aliquots of calcite 246 powder were reacted with anhydrous (103%) phosphoric acid in a Kiel IV carbonate device. The resulting 247 CO<sub>2</sub> was cryogenically purified and cleaned using a PoraPak Q trap kept at -40 °C (Petersen et al., 2016b) 248 before being led into a MAT253 or MAT253 PLUS IRMS via a Dual Inlet system. Intensities on masses 44-249 49 of the CO₂ samples were measured using the Long Integration Dual Inlet mode (Müller et al., 2017) 250 with 400 s integration time against a clean CO<sub>2</sub> working gas ( $\delta^{13}$ C = -2.82 %;  $\delta^{18}$ O = -4.67 %), corrected 251 for the pressure baseline (He et al., 2012). Clumped isotope values ( $\Delta_{47}$ ) were brought into the Intercarb 252 Carbon Dioxide Equilibrium Scale (I-CDES; (Bernasconi et al., 2021) using the three ETH standards (ETH-1, 253 ETH-2 and ETH-3) and their accepted values (Bernasconi et al., 2018). Throughout this procedure, samples 254 were corrected with standards whose signal intensities on the mass 44 cup deviated from the intensities 255 of the samples by less than 1 V to prevent any intensity-based offset in the clumped isotope values to bias 256 the result. For samples for which less than 5 intensity-matched standards were available for this correction, 257 the  $\Delta_{47}$  value was not considered in the rest of the analysis. The complete analytical system was monitored 258 regarding performance with two independent standards (IAEA-C2, N = 49, and Merck, N = 48), which were 259 treated as samples throughout the measurement procedure. The standard deviations of  $\Delta_{47}$  values of 260 these check standards were 0.053% for IAEA-C2 and 0.039 % for Merck. The reproducibility standard 261 deviations of  $\delta^{13}$ C and  $\delta^{18}$ O for IAEA-C2 were 0.05 ‰ and 0.09 ‰ respectively and for Merck the reproducibility standard deviations were 0.09 % and 0.13 % for  $\delta^{13}$ C and  $\delta^{18}$ O respectively. Results of 262 263 clumped and associated stable isotope data are reported in **Supplement S2** in the Zenodo repository. A 264 summary of the  $\delta^{13}$ C,  $\delta^{18}$ O and  $\Delta_{47}$  values organized by the four regions of **HU-027** that were sampled (see 265 Figure 2) is presented in Table 2.

**Table 2**: Statistics of carbon, oxygen and clumped isotope results organized per sampled region in specimen **HU-027**.

| Sampled region:    | F                             | R_01-R_0                      | 7                             | F                             | _08-R_1                       | 1                             |                 | RB_1                          |                               | RB_2                     |                               |                               |  |  |
|--------------------|-------------------------------|-------------------------------|-------------------------------|-------------------------------|-------------------------------|-------------------------------|-----------------|-------------------------------|-------------------------------|--------------------------|-------------------------------|-------------------------------|--|--|
| Proxy:             | Δ <sub>47</sub><br>(‰ I-CDES) | δ <sup>18</sup> O<br>(‰ VPDB) | δ <sup>13</sup> C<br>(‰ VPDB) | Δ <sub>47</sub><br>(‰ I-CDES) | δ <sup>18</sup> O<br>(‰ VPDB) | δ <sup>13</sup> C<br>(‰ VPDB) | Δ <sub>47</sub> | δ <sup>18</sup> O<br>(‰ VPDB) | δ <sup>13</sup> C<br>(‰ VPDB) | $\Delta_{47}$ (% I-CDES) | δ <sup>18</sup> O<br>(‰ VPDB) | δ <sup>13</sup> C<br>(‰ VPDB) |  |  |
| N                  | 29                            | 35                            | 35                            | 16                            | 20                            | 20                            | 27              | 27                            | 27                            | 14                       | 14                            | 14                            |  |  |
| Median value       | 0.544                         | -6.12                         | 0.66                          | 0.560                         | -4.71                         | 2.02                          | 0.608           | -5.81                         | 1.37                          | 0.569                    | -5.02                         | 1.67                          |  |  |
| Mean value         | 0.543                         | -6.13                         | 0.73                          | 0.560                         | -4.71                         | 2.02                          | 0.614           | -5.80                         | 1.37                          | 0.577                    | -5.03                         | 1.67                          |  |  |
| Standard deviation | 0.032                         | 0.26                          | 0.35                          | 0.035                         | 0.20                          | 0.12                          | 0.037           | 0.06                          | 0.04                          | 0.027                    | 0.08                          | 0.02                          |  |  |
| Minimum            | 0.460                         | -6.75                         | -0.14                         | 0.504                         | -5.22                         | 1.76                          | 0.557           | -5.93                         | 1.30                          | 0.540                    | -5.28                         | 1.60                          |  |  |
| Maximum            | 0.589                         | -5.67                         | 1.43                          | 0.619                         | -4.42                         | 2.28                          | 0.697           | -5.70                         | 1.46                          | 0.621                    | -4.97                         | 1.70                          |  |  |

#### 2.3 Modern bivalve occurrence and climate

To contrast the paleoclimate reconstructions from the above-mentioned multi-proxy dataset, we extracted data on the occurrences of bivalves in modern oceans from the Ocean Biodiversity Information System database (OBIS, 2020). We accessed the OBIS database through the "occurrences" function of the "robis" package (Provoost et al., 2022) and processed the occurrences in the open-source computational software package R (R Core Team, 2023). A total of 2199523 occurrences of taxa in the class of Bivalvia were extracted from the database, and their localities were categorized in bins of 2° by 2° based on their latitude and longitude.

Modern seasonal SST ranges across the world oceans were extracted from the Extended Reconstructed Sea Surface Temperature (ERSST) dataset from the National Oceanic and Atmospheric Administration (NOAA; (Huang et al., 2017)). We extracted monthly data from the years 1981 until 2010 on a 2° by 2° latitude and longitude grid from the ERSST dataset. We extracted the warmest and coldest monthly temperatures from the grid cells that contained reported occurrences of bivalves in the OBIS dataset. This resulted in a distribution of the warmest and coldest monthly temperatures across the living environment of modern bivalves.

### 2.4 Data processing

Our clumped isotope dataset on specimen **HU-027** yielded information about the seasonal spread in temperature and the water oxygen isotopic value ( $\delta^{18}O_w$ ) experienced by this specimen. We reconstructed these parameters from the  $\Delta_{47}$  and  $\delta^{18}O_c$  values of the shell carbonate following the clumped isotope-temperature relationship by (Daëron and Vermeesch, 2023) and the calcite  $\delta^{18}O_c$ - $\delta^{18}O_w$ -temperature relationship by (Kim and O'Neil, 1997). This dataset allowed us to characterize seasonal variability in climate in the Saiwan paleoenvironment. However, since the relatively short  $\delta^{18}O$  profile from **HU-027** did not allow age modelling and the clumped isotope dataset only sampled one specimen, we augmented the dataset from **HU-027** with  $\delta^{18}O$ ,  $\delta^{13}C$  and trace element data from other specimens (see section 2.2.3) in the same assemblage to study the relationship between temperature and growth rate in Saiwan. To achieve this, we carried out subsequent data processing steps outlined below.

To reconstruct shell growth rates of the molluscs and water temperatures in the Saiwan environment from chemical data measured in the shells, we carried out the following data processing steps (see also the flowchart in **Figure 3**):

- 1. We used concentrations of Mn and Fe and Mg/Ca and Sr/Ca ratios to screen for diagenetic recrystallization of parts of the shells, and remove chemical data from suspicious shell sections for further analysis.
- 2. We applied the ShellChron age model (de Winter, 2021) to produce internal shell chronologies based on seasonal cyclicity in  $\delta^{18}$ O and  $\delta^{13}$ C values through each shell profile.
- 3. We applied the Daydacna age model (Arndt et al., 2023) to produce internal shell chronologies based on subdaily-scale trace element variability in specimen **B10** to verify the result of the ShellChron algorithm.
- 4. We use a combination of oxygen isotope and clumped isotope data, grouped per location in specimen **HU-027** to reconstruct seasonal changes in temperature and  $\delta^{18}O_w$  in the Saiwan environment, and how they relate to the oxygen isotope variability in the shells.
- 5. We use the information about  $\delta^{18}O_w$  variability in the environment from clumped isotope data in **HU-027** to reconstruct seasonal temperature variability in the Saiwan ecosystem based on all  $\delta^{18}O$  profiles.
- 6. We combine seasonal-scale information about shell growth rates and temperatures to determine the maximum temperature at which the mollusc species mineralized their shell and to quantify the effect of temperature on shell growth rates in Saiwan.

Details on the diagenetic screening are discussed in **sections 3.2** and **4.1**. Assumptions and details regarding all data processing steps are explained below.

**Figure 3:** Flowchart of data processing steps carried out for this study.

2.4.1 Seasonal scale age models using ShellChron

Internal age models were created for all  $\delta^{18}$ O and  $\delta^{13}$ C profiles in all specimens except for **HU-027**, for which the  $\delta^{18}$ O and  $\delta^{13}$ C profile was too short to meet the criteria for applying the algorithm (see below),

as well as the Mg/Ca and Sr/Ca profiles in *T. sanchezi* specimen **B10** based on the growth rate modelling routine ShellChron (de Winter, 2021). ShellChron approximates the shape of the proxy curve from combinations of sinusoidal proxy and growth rate curves. The routine was adapted after the work by (Judd et al., 2017) to function in a sliding window algorithm to provide one age-distance model for the entire profile, preventing breaks and time jumps between growth years. ShellChron can approximate the internal chronology of any proxy-distance record using the following assumptions:

- 1. The proxy has a (quasi) periodic behaviour over the year. In other words: The proxy exhibits one maximum and one minimum per annual cycle.
- 2. The mineralization (or growth) rate of the archive over a year can be approximated by a (skewed) sinusoid, with one annual maximum and one minimum (which can be zero).
- 3. The proxy record contains at least 2 full annual cycles to allow for sufficient overlap between moving windows.

ShellChron estimates the uncertainties of age estimates per datapoint by comparing the results of overlapping windows on the proxy record during the sliding window approach which is applied in the model. Since the proxy-depth record in each window in the ShellChron model is estimated separately using a new combination of proxy and growth rate sinusoid, subsequent age estimates for the same distance value can have different, independent outcomes with respect to relative age estimate. In addition, ShellChron propagates the uncertainty on the distances and proxy values (if provided) using a Monte Carlo approach, resulting in a realistic estimate of the uncertainty on the age determination (de Winter, 2021). For each datapoint, the model thus produces a distribution of ages, from which an uncertainty on the relative age of each datapoint is obtained. These can in turn be averaged to gauge the overall precision of the model outcome. In addition, wide age distributions for the same datapoint are indicative of misidentifications of annual cycles in the  $\delta^{18}$ O profile, which cause bifurcations in the model outcome, increasing the spread in age outcomes. Therefore, the overall precision of the ShellChron outcome yields information about the certainty of age modelling and can be used as a benchmark for selecting the most reliable age-distance relationship in a shell. The shell height vs age relationships estimated using ShellChron for multiple proxy records ( $\delta^{18}$ O,  $\delta^{13}$ C and trace element ratios, if available) from each specimen except HU-027, including their uncertainties, are provided in Supplement S3 in the Zenodo repository. The age-distance relationships resulting from the most precise age model for each specimen were used to assign a time of the year to each stable isotope and trace element datapoint used in this study.

### 2.4.2 Sub-seasonal age model using Daydacna

For *T. sanchezi* specimen **B10**, from which subdaily-scale Mg/Ca, Mg/Li, Sr/Ca and Sr/Li data were available (de Winter et al., 2020), we applied the growth modelling routine Daydacna (Arndt et al., 2023) to verify the ShellChron results and enhance the resolution of the age model. Daydacna uses a wavelet transformation to detect daily rhythms in chemical profiles through mollusc shells and applies a userguided peak identification routine to find age-depth relationships in the shell on a daily scale. We applied the Daydacna routine on subdaily scale Mg/Ca, Mg/Li, Sr/Ca and Sr/Li records one by one through specimen **B10** (see script in **Supplement S4** in the Zenodo repository). We compare the results of Daydacna with the results of ShellChron, which are based on annual cycles in  $\delta^{18}$ O and  $\delta^{13}$ C profiles and are therefore independent from the Daydacna results based on daily cycles in trace element ratios.

## 2.4.3 Monthly binning of isotope data

366367

372373

Stable isotope data from profiles through all specimens were cross-referenced with shell age results from ShellChron and Daydacna. Datapoints for which age modelling did not yield a conclusive age (e.g. stable isotope datapoints from locations in between data in trace element profiles) were dated by linear interpolation between surrounding samples for which dates were available. All chemical data were then binned into monthly time bins based on the age models. Monthly bins were assigned by dividing the year into 12 equal time segments, defining boundaries between months based on the day of the year. These monthly time bins were assigned 30 times for each specimen, shifting the boundaries between the months by 1 modelled day for each new assignment. The optimal monthly assignment was subsequently found by picking the option out of 30 in which the months with the highest and lowest mean  $\delta^{18}$ O value exhibited the highest difference. This age assignment is assumed to find the highest (least smoothed) seasonal variability in  $\delta^{18}$ O, while staying true to the sub-annual growth rate variability exhibited by the specimen as modelled by the ShellChron and Daydacna algorithms.

- Note that this monthly binning assumes a total of 365 days in a year, while in reality the number of days per year during the Late Cretaceous was higher (de Winter et al., 2020). In addition, growth stops may occur which prevent the mollusc from recording one or more days during periods of stress (Jones, 1983), even though no clear signs of these were directly observed in our specimens. However, this difference does not influence the monthly binning since 12 equal parts of the year were considered and the number of days per year assumed in the ShellChron and Daydacna models was also set to 365.
- 2.4.4 Combining clumped isotope and  $\delta^{18}$ O data
- To overcome the lack of precise intra-shell age control in **HU-027** and place the clumped isotope results in a seasonal context, we grouped the  $\Delta_{47}$  and carbonate  $\delta^{18}O$  ( $\delta^{18}O_c$ ) data from specimen **HU-027** in four bins according to their sampling location (see **Figure 2** and **Table 2**). This resulted in a seasonal spread of  $\Delta_{47}$ ,  $\delta^{18}O_c$  and  $\delta^{13}C$  values which allowed us to quantify the relationship between  $\delta^{18}O_c$  and  $\Delta_{47}$  (and therefore temperature and  $\delta^{18}O_w$ ) in the Saiwan environment.
- To verify whether our choice of sampling locations for clumped isotope analysis in specimen HU-027 387 388 sampled the full seasonal spread in (clumped) isotopic values, we applied a clustering routine to the 389 carbonate  $\delta^{18}$ O ( $\delta^{18}$ O<sub>c</sub>) and  $\delta^{13}$ C values of **HU-027** using K-means and Partitioning Around Medioids (PAM) 390 clustering routines (see Supplement S5 in the Zenodo repository). The K-means routine groups datapoints 391 in the  $\delta^{18}O_c$ - $\delta^{13}C$  space into clusters minimizing the squared Euclidian distance between the points within 392 a cluster using the iterative Hartigan-Wong algorithm coded in the "kmeans" function of the "stats" package in R (Hartigan and Wong, 1979; R Core Team, 2023; "stats package," 2019). Clustering was 393 394 repeated on the same dataset using the PAM algorithm (Kaufman and Rousseeuw, 1990) using the "pam" 395 function of the "cluster" package ("cluster package," 2023; Maechler et al., 2023).
- Because of the strong seasonal cycles in productivity, dissolved inorganic carbon composition, freshwater influx and temperature in shallow marine settings, summer and winter seasons are typically recorded through distinct combined  $\delta^{18}O_c$  and  $\delta^{13}C$  signatures in mollusc shells (De Winter et al., 2018; McConnaughey and Gillikin, 2008; Surge et al., 2001). By combining the statistical clustering approach on  $\delta^{18}O_c$  and  $\delta^{13}C$  data with this knowledge of typical seasonal isotopic signatures, we verified the assignment of summer and winter seasons in the geochemical record of **HU-027** independent from their sampling location. Note that the assignment of bins in our clumped isotope dataset does not rely on this clustering

outcome, as the binning was based primarily on location in the shell and only cross-checked with the statistical clustering.

We modelled the relationship between  $\delta^{18}O_c$ ,  $\delta^{18}O_w$  and clumped isotope-based temperature in the Saiwan environment from the data in specimen **HU-027**. To do so, we used the relative timing of the four clumped isotope clusters (RB\_1 directly preceding R\_8 – R\_11 and RB\_2 preceding and partly overlapping with R\_1 – R\_7; see **Figure 2**) and the assumption that the warmest and coldest clusters record summer and winter temperatures, respectively, to determine the order of the clumped isotope clusters throughout the year. We then simulated the pathways between consecutive clusters in the  $\delta^{18}O_c$ ,  $\delta^{18}O_w$  and temperature through a Monte Carlo simulation, taking into account the uncertainty on these three parameters within the clusters. We simulated 1000 linear pathways between the clusters consisting of 100 steps while preserving the relationships between  $\delta^{18}O_c$ ,  $\delta^{18}O_w$  and temperature, sampling the start and end points from the uncertainty distributions of the parameters in the clusters.

We then estimated  $\delta^{18}O_w$  values for each stable isotope measurement in our compilation for which no clumped isotope values were available using this seasonal  $\delta^{18}O_c$ - $\delta^{18}O_w$  relationship. Since the cyclical nature of the seasonal  $\delta^{18}O_c$ - $\delta^{18}O_w$  relationship produces non-unique  $\delta^{18}O_w$  estimates for any given  $\delta^{18}O_c$  value (see section 3.4), we used the seasonal timing of the  $\delta^{18}O_c$  datapoints to obtain the most likely  $\delta^{18}O_w$  outcome:  $\delta^{18}O_c$  values associated with the warm, high- $\delta^{18}O_w$  spring/summer season (before the summer  $\delta^{18}O_c$  minimum) were assigned the highest of the two possible  $\delta^{18}O_w$  outcomes. Samples associated with the lower temperature and  $\delta^{18}O_w$  half of the seasonal cycle (after the summer  $\delta^{18}O_c$  minimum) were assigned the lowest of the two possible  $\delta^{18}O_w$  outcomes. For the specimens of V vesiculosus (B6) and O. figari (B11), which exhibit higher  $\delta^{18}O_c$  values than the T. sanchezi specimens,  $\delta^{18}O_w$  values were assigned based on the seasonal timing of the  $\delta^{18}O_c$  values. Temperatures were calculated for all  $\delta^{18}O_c$  outcomes in our compilation based on the  $\delta^{18}O_c$  measurements and  $\delta^{18}O_w$  estimates. We produce monthly temperature and  $\delta^{18}O_w$  estimates for each specimen by grouping the data obtained from applying the clumped isotope-derived  $\delta^{18}O_c$ - $\delta^{18}O_w$ -temperature relationship on  $\delta^{18}O_c$  profiles in monthly bins per specimen.

To test the sensitivity of our  $\delta^{18}O_w$  and temperature estimates to the observation that the  $\delta^{18}O_w$  varies seasonally following the **HU-027** clumped isotope outcomes, we also calculated temperatures for all  $\delta^{18}O_c$  profiles using a constant  $\delta^{18}O_w$  value equal to the mean  $\delta^{18}O_w$  value of the three warmer clusters identified in the clumped isotope dataset from **HU-027** (average: -0.25 %VSMOW), excluding the cluster that has a low (-4.61  $\pm$  0.86 %VSMOW)  $\delta^{18}O_w$  value. We repeated this test assuming the classical (and seasonally constant)  $\delta^{18}O_w$  value of -1 % VSMOW, which is often thought to represent fully marine conditions in a land ice-free climate (Shackleton, 1986). We discuss the impact of the decision not to consider seasonal variability in  $\delta^{18}O_w$  values for these specimens in **section 4.3.** 

### **3. Results**

## 3.1 Stable isotope results

All specimens show distinct periodic patterns in  $\delta^{18}O_c$  and  $\delta^{13}C$  values when plotted in growth direction through the shells (**Figure 4**). Stable oxygen isotope values ( $\delta^{18}O_c$ ) in the shells vary between -7.1 %-VPDB and -1.5 %-VPDB with a mean  $\delta^{18}O_c$  value of -5.1  $\pm$  0.9 %-VPDB (1 $\sigma$ ) for the entire dataset. The lowest mean  $\delta^{18}O_c$  values are recorded in *T. sanchezi* shells (-5.5  $\pm$  0.6 %-VPDB; 1 $\sigma$ ), with higher values recorded in *V. vesiculosus* (-4.1  $\pm$  0.6 %-VPDB; 1 $\sigma$ ) and *O. figari* (-3.6  $\pm$  0.4 %-VPDB; 1 $\sigma$ ). Similarly, the lowest mean  $\delta^{13}C$  values are recorded in *T. sanchezi* (0.7  $\pm$  0.9 %-VPDB; 1 $\sigma$ ), followed by *V. vesiculosus* (0.8  $\pm$  0.4 %-VPDB; 1 $\sigma$ ) and *O. figari* (1.7  $\pm$  0.6 %-VPDB; 1 $\sigma$ ).

**Figure 4**: Overview of 11 incrementally sampled stable oxygen ( $\delta^{18}O_c$ ; blue) and carbon isotope ( $\delta^{13}C$ ; red) profiles: 9 profiles through 5 *T. sanchezi* specimens, of which 5 parallel profiles through specimen **H579** (**A, B, C, D & F**; green frame), one profile through *V. vesiculosus* specimen **B6** (**E**; purple frame) and one profile through *O. figari* specimen **B11** (**G**; orange frame). Vertical axes of *T. sanchezi* profiles in **A-D** are equal, while *V. vesiculosus*, *O. figari* and *T. sanchezi* specimen **HU-027** have different vertical axes. Records in **D** represent parallel profiles through the same specimen (**H579**). The shaded background colours represent time of year based on the ShellChron chronologies constructed using these  $\delta^{18}O_c$  and  $\delta^{13}C$  profiles, with darker colour indicating samples assigned to days earlier in the year. Black dots in **G** show  $\delta^{18}O_c$  and  $\delta^{13}C$  values associated with clumped isotope measurements in **HU-027** and coloured dots and error bars indicate the spread in  $\delta^{18}O_c$  and  $\delta^{13}C$  and the location of the material used in clumped clusters presented in **Figure 7**.

### 3.2 Trace element results

Trace element analyses highlight that shells of all three species are generally characterized by low concentrations of Mn and Fe (**Figure 5**). *T. sanchezi* exhibits median Mn concentrations of 38  $\mu$ g/g (average: 50 ± 44  $\mu$ g/g, 1 $\sigma$ ) and median Fe concentrations of 56  $\mu$ g/g (average: 77 ± 116  $\mu$ g/g, 1 $\sigma$ ) with a few isolated locations in the shell with concentrations exceeding 300  $\mu$ g/g and 1000  $\mu$ g/g for Mn and Fe,

respectively. The carbonate in *V. vesiculosus* has somewhat higher median Mn concentrations of 176  $\mu$ g/g (average: 192 ± 94  $\mu$ g/g, 1 $\sigma$ ) and median Fe concentrations of 125  $\mu$ g/g (average: 173 ± 152  $\mu$ g/g, 1 $\sigma$ ) with some locations showing Mn and Fe concentrations exceeding 500  $\mu$ g/g and 800  $\mu$ g/g, respectively. A clear positive trend is observed towards higher Mn and Fe concentrations in *V. vesiculosus* samples (**Fig. 5A**). Finally, *O. figari* has median Mn concentrations of 63  $\mu$ g/g (average: 75 ± 33  $\mu$ g/g, 1 $\sigma$ ) and much lower median Fe concentrations of 4  $\mu$ g/g (average: 5.7 ± 5.8  $\mu$ g/g, 1 $\sigma$ ), with maximum Mn and Fe concentrations of 180  $\mu$ g/g and 40  $\mu$ g/g, respectively, in some locations.

Mg/Ca and Sr/Ca ratios are very similar between T. sanchezi and V. vesiculosus, with mean Mg/Ca ratios of 11.4  $\pm$  2.4 mmol/mol (1 $\sigma$ ) for T. sanchezi and 11.5  $\pm$  4.8 mmol/mol (1 $\sigma$ ) for V. vesiculosus and mean Sr/Ca ratios of 1.49  $\pm$  0.21 mmol/mol (1 $\sigma$ ) for T. sanchezi and 1.09  $\pm$  0.45 mmol/mol (1 $\sigma$ ) for V. vesiculosus. A subset of the samples from V. vesiculosus exhibit a clear trend towards lower Mg/Ca and Sr/Ca values. O. figari exhibits much lower Mg/Ca values (1.87  $\pm$  1.15 mmol/mol; 1 $\sigma$ ) and Sr/Ca values of 1.36  $\pm$  0.18 mmol/mol (1 $\sigma$ ).

**Figure 5:** Cross plots of manganese vs iron (**A**) and magnesium vs strontium (**B**) concentrations in *T. sanchezi* (green), *V. vesiculosus* (purple) and *O. figari* (orange) measured using micro-XRF line scan analysis. Shaded points highlight individual measurements in profiles through the shells while bold black crossed lines highlight median concentration values with 2 standard deviations of the variability per species. Histograms on the edges of the plot show the distribution of concentration values in the dataset per element.

### 3.3 Age model results

Applying ShellChron on Mg/Ca, Sr/Ca (only for specimen **B10**),  $\delta^{18}O_c$  and  $\delta^{13}C$  values through all specimens except **HU-037** yielded information about the age-distance relationship in the direction of growth through the shells (**Figure 6**). These distances in growth direction on cross-sections through the shells were interpreted as proxies for the growth rate of the individual during its life. ShellChron-based age models

yield highly consistent age-distance relationships for different *T. sanchezi* specimens, regardless of whether they are based on Mg/Ca, Sr/Ca,  $\delta^{18}O_c$  or  $\delta^{13}C$  records (**Figure 6A**). Contrarily, growth models based on  $\delta^{18}O_c$  and  $\delta^{13}C$  values in *V. vesiculosus* and *O. figari* differed (**Figure 6B-C**).

Applying the Daydacna algorithm to trace element records through *T. sanchezi* specimen **B10** (for which subdaily-resolved trace element data is available) yielded independent evidence for the age-depth relationship in shells of *T. sanchezi* (see **Figure 6D**). Except for the Mg/Li record, all age-distance relationships obtained by applying Daydacna on trace element records through specimen **B10** closely agree with the age-distance relationship obtained through the combined ShellChron growth models for this and other *T. sanchezi* specimens. (**Figure 6B**). Of the Daydacna results, the model based on Mg/Li ratios deviates most strongly from the other Daydacna results (based on Mg/Ca, Sr/Ca and Sr/Li records) and the stable isotope-based ShellChron age models. The close agreement between these age models generated using different algorithms, based on different environmental cycles (daily vs seasonal) on different geochemical records through the same specimen, highlights the reproducibility of the age-distance relationship found for our assemblage of *T. sanchezi* specimens from the Saiwan ecosystem. This lends confidence to the interpretation that the observed rhythms in Mg/Ca, Sr/Ca and Sr/Li records in specimen **B10** represent daily cycles (de Winter et al., 2020) and allows us to refine our age model for this species to quantify growth rates on a monthly scale.

Isotope profiles in **Figure 4** and data on the precision of the ShellChron model outcomes in **Table 3** show that the seasonal pattern is in some specimens clearer in  $\delta^{18}O_c$ , while others show clearer seasonality in the  $\delta^{13}C$  records. This translates to a better precision of  $\delta^{18}O_c$ -based age models in some specimens, while others have better-defined age models based on  $\delta^{13}C$ . On average, the  $\delta^{18}O_c$ -based age models are more precise (23.1 days at 95% confidence level) than the  $\delta^{13}C$ -based age models (27.6 days at 95% confidence level; see **Table 3**). To account for inter-specimen differences, we decided to use the most precise growth model (either based on  $\delta^{18}O_c$  or  $\delta^{13}C$ ) available per specimen for determining the age-distance relationship.

**Table 3**: Median uncertainty (95% CL; in days) of growth models based on  $\delta^{18}O_c$  and  $\delta^{13}C$  profiles through all sequentially sampled specimens. For each specimen, the most precise age model is highlighted in bold.

| Specimen | Species        | $\delta^{18}O_c$ -based model | $\delta^{\text{13}}\text{C-based model}$ |
|----------|----------------|-------------------------------|------------------------------------------|
|          |                | uncertainty (days)            | uncertainty (days)                       |
| B6       | V. vesiculosus | 22.9                          | 17.7                                     |
| B11      | O. figari      | 24.7                          | 57.4                                     |
| B10      | T. sanchezi    | 22.0                          | 11.3                                     |
| H576     | T. sanchezi    | 27.7                          | 32.3                                     |
| H579A    | T. sanchezi    | 18.4                          | 40.4                                     |
| H579B    | T. sanchezi    | 19.4                          | 39.2                                     |
| H579C    | T. sanchezi    | 13.4                          | 10.2                                     |
| H579D    | T. sanchezi    | 32.0                          | 27.2                                     |
| H579E    | T. sanchezi    | 21.4                          | 9.3                                      |
| H585     | T. sanchezi    | 29.0                          | 31.0                                     |
| AVERAGE  |                | 23.1                          | 27.6                                     |

519

529

Figure 6: Plot of shell height vs. age in all specimens based on ShellChron modelling on Mg/Ca, Sr/Ca,  $\delta^{18}$ O<sub>c</sub> and  $\delta^{13}$ C profiles in *T. sanchezi* (A),  $\delta^{18}$ O<sub>c</sub> and  $\delta^{13}$ C profiles in *V. vesiculosus* (B) and  $\delta^{18}$ O<sub>c</sub> and  $\delta^{13}$ C profiles in O. figari (C). Solid lines represent LOESS smoothed curves (span = 0.2) through the age-height data, with shaded areas indicating uncertainties around the growth models. D) Comparison between ShellChron results for T. sanchezi specimen **B10** profiles (combined chronology from all proxies; black line in panel A) and Daydacna results on subdaily-scale trace element profiles through the same specimen. Colours of curves and uncertainty envelopes represent the proxy on which age modelling was based (see legends in top-left corners of the panels).

## 3.4 Clumped isotope results

Clumped isotope analysis on *T. sanchezi* specimen **HU-027** yielded a mean  $\Delta_{47}$  value of 0.572  $\pm$  0.047 % I-CDES (10). The clusters created from the clumped isotope dataset of specimen HU-027 highlight the relationship between  $\delta^{18}O_c$  values in *T. sanchezi* and the temperature and  $\delta^{18}O_w$  values reconstructed from clumped isotope thermometry (Figure 7). Clustering by location in the shell yields maximum reconstructed temperatures in HU-027 of 44.2 ± 4.0°C and minimum temperatures of 19.2 ± 3.8°C. These clusters in HU-**027** sample a similar or larger spread in  $\delta^{18}O_c$ ,  $\delta^{13}C$  and  $\Delta_{47}$  compared to the statistical clustering approaches that do not take into account the sample location (maximum temperature range: 45.4 ± 17.1°C to  $24.7 \pm 4.3$ °C; see **Supplement S5** in the Zenodo repository), demonstrating that the sampling strategy successfully resolves the seasonal variability recorded in *T. sanchezi* specimen **HU-027**.

Interestingly, while the lowest  $\delta^{18}O_c$  values in T. sanchezi are associated with the highest temperatures, as one would expect assuming a constant  $\delta^{18}O_w$  value throughout the year, the highest  $\delta^{18}O_c$  values do not represent the coldest season. This suggests that the Saiwan environment in the Late Campanian experienced significant seasonal variability in  $\delta^{18}O_w$  values. Combining clumped and oxygen isotope data on the clusters shows indeed that they record excursions towards very low  $\delta^{18}O_w$  values (-4.63  $\pm$  0.86 %VSMOW) in the coldest season, far off the  $\delta^{18}O_w$  value of -1 %VSMOW commonly assumed for past greenhouse periods (Shackleton, 1986).

**Figure 7**: Relationship between temperature and  $\delta^{18}O_c$  values (**A**) and  $\delta^{18}O_w$  and  $\delta^{18}O_c$  values (**C**) in *T. sanchezi* shell based on four clusters of clumped isotope analyses through specimen **HU-027** grouped by location of the samples in the shell cross-section (see symbol legend on top). Uncertainties on mean cluster values are reported as 95% confidence levels. The thin black lines and grey shading highlight individual Monte Carlo simulations (N = 1000) of the most likely shape of the  $\delta^{18}O_c$ -temperature and  $\delta^{18}O_c$ - $\delta^{18}O_w$  relationship and their 68% and 95% confidence levels. The horizontal dashed line in **B** indicates the common assumption of a constant  $\delta^{18}O_w$  value of -1 %VSMOW throughout the year in the land ice-free Late Cretaceous. Plots **B** and **D** show an interpretation of the seasonality in temperature and  $\delta^{18}O_w$  in the Saiwan environment based on these clusters.

### 4. Discussion

## 4.1 Shell preservation

Ancient shell carbonates, such as the rudists studied here, are known to be susceptible to various forms of diagenetic alteration (Al-Aasm and Veizer, 1986a, 1986b; Brand and Veizer, 1981, 1980; Ullmann and Korte, 2015). Common in carbonate systems, such as the carbonate platforms where the Saiwan rudists were growing, is open-system diagenesis, in which the diagenetic fluid with which the shell carbonate exchanges to alter its chemical and isotopic composition is continuously replaced (Al-Aasm and Veizer, 1986a; Brand and Veizer, 1981). In such systems, the chemical and isotopic composition of carbonates moves away from its original value in an approximately linear trend (mixing line), typically resulting in increased concentrations of trace elements such as Mn and Fe, reduced concentrations of Sr and trends towards lower  $\delta^{18}O_c$  values in the carbonate (Al-Aasm and Veizer, 1986a, 1986b).

Our detailed geochemical investigation of specimens **B6**, **B10** and **B11** (**Figure 2**; **Figure 5**) highlights low Mn and Fe concentrations (typically <300  $\mu$ g/g and <200  $\mu$ g/g, respectively, below thresholds used by (Schmitt et al., 2022); see **Figure 5**) and high Sr concentrations (typically >1.0 mmol/mol; see **section 3.2**), which show a correlated, skewed distribution tailing towards a few locations on the shell where Mn and Fe concentrations are high and Sr concentrations are low, especially in *V. vesiculosus* specimen **B6** (**Figure 5**). In *V. vesiculosus*, the trend of covarying elevated Mn and Fe concentrations and coinciding reductions in Mg/Ca and Sr/Ca clearly shows the imprint of local open-system diagenetic remineralization. Open system diagenesis was observed in isolated localities in these fossil shells, which were avoided during sampling for stable and clumped isotope analysis (see **section 2.2.3**). The chemical differences between well-preserved sections of the shells of these three taxa are likely to reflect taxon-specific variations in trace element concentrations, which are also common in modern molluscs and may reflect differences in shell microstructures and their associated formation pathways (e.g. Carré et al., 2006; Onuma et al., 1979).

Isotopic compositions of carbon and oxygen are often jointly depleted in diagenetically altered materials due to the exchange of shell carbonate with either isotopically depleted meteoric fluids during early diagenetic alteration (Allan and Matthews, 1990) or exchange with pore fluids under high temperatures (e.g. Brand and Veizer, 1981). However, a positive correlation between  $\delta^{18}O_c$  and  $\delta^{18}C$  values is not necessarily a reliable indicator of diagenetic alteration (Swart and Oehlert, 2018). Instead, a positive correlation between  $\delta^{18}O_c$  and  $\delta^{18}C$  values is often observed in modern (non-diagenetically altered) photosymbiotic species, such as tridacnids (Elliot et al., 2009; Killam et al., 2020). Such a correlation has been proposed to be caused by seasonal changes in the isotopic composition of the dissolved inorganic carbon pool due to variability in the activity of photosymbionts in phase with the seasonal effect of temperature on the oxygen isotope composition of the shell (Elliot et al., 2009; McConnaughey and Gillikin, 2008). The fact that T. sanchezi specimens in our dataset exhibit a strong positive correlation between  $\delta^{18}O_c$  and  $\delta^{18}C$ , while the other species do not, is corroborated by other evidence that *T. sanchezi* had photosymbionts such as the presence of specific adaptations in the shell thought to facilitate the hosting of photosymbiotic microorganisms in the mantle and the strong expression of diurnal cycles in shell structure and chemistry (see (N. J. de Winter et al., 2020; Skelton and Wright, 1987; Steuber, 1999)). We therefore disregard this as evidence for open-system diagenesis.

Grain boundary diffusion is another potential diagenetic process that can influence the isotopic composition of biogenic carbonates. This process is rapid on geological timescales (<100 years) and causes the exchange of oxygen isotopes with pore fluids (Adams et al., 2023; Cisneros-Lazaro et al., 2022;

Nooitgedacht et al., 2021). In foraminifera, this process exchanges up to ~3% of the oxygen in the biomineral (Adams et al., 2023), meaning that even in the presence of strongly isotopically negative pore fluids this process can only change the  $\delta^{18}$ O<sub>c</sub> value of the biomineral by a few tens of a permille, not enough to fully explain the high temperatures recorded in the fossils in this study.

Finally, our clumped isotope analysis results of specimen **HU-027** may be susceptible to solid-state reordering of the clumped isotope signature, a form of closed-system diagenesis which occurs at elevated temperatures (>100 °C; Chen et al., 2019; Looser et al., 2023; Passey and Henkes, 2012; Stolper and Eiler, 2015). This reordering effect requires large differences between the temperatures in which the carbonate was originally precipitated and the temperatures of the surrounding rocks and, when activated, is likely to affect the entire sample equally rapidly (on geological timescales; Henkes et al., 2014; Stolper and Eiler, 2015). Similarly, heating carbonate samples to 175°C caused a resetting of the  $\Delta_{47}$  value through exchange with internal waters without noticeable change to  $\delta^{18}O_c$  values (Nooitgedacht et al., 2021). Given the fact that significant temperature variability is recorded by the clumped isotope dataset from specimen **HU-027**, and that the recorded temperatures are far from the temperatures needed for the solid-state reordering process to significantly affect isotopic clumping (>100 °C; Henkes et al., 2014), we feel confident in interpreting the recorded temperatures in terms of the paleoclimate and environment at the Saiwan site.

# 4.2 Seasonality in temperature and $\delta^{18}O_w$ value in Saiwan

The Monte Carlo simulations of the seasonal  $\delta^{18}O_{c}$ - $\delta^{18}O_{w}$ -temperature path based on the clumped isotope dataset from specimen HU-027 in Figure 7 show a statistically significant difference between paleotemperatures and  $\delta^{18}O_w$  values between two parts of the annual cycle, especially for the middle range of the  $\delta^{18}O_c$  values of ( $\delta^{18}O_c$  between -5.0 and -6.0 %VPDB). Shell increments deposited after the  $\delta^{18}O_c$  minimum record lower temperature and  $\delta^{18}O_w$  values, while parts of the shell before the  $\delta^{18}O_c$ minimum record high temperature and  $\delta^{18}O_w$  values. High temperatures are reconstructed for both the extreme ends of the  $\delta^{18}O_c$  variability ( $\delta^{18}O_c < -6.0$  and  $\delta^{18}O_c > -5.0$  %VPDB). We interpret this as the signature of a shift in the temperature and  $\delta^{18}O_w$  maxima with respect to the  $\delta^{18}O_c$  cycle, with a high temperature extreme (44.2  $\pm$  4.0°C) at the low end of the  $\delta^{18}O_c$  cycle, which we interpret as a hot and dry summer season, and a milder temperature maximum (41.4  $\pm$  4.8°C) at the high end of the  $\delta^{18}O_c$  cycle, which we interpret as a warm and dry spring season (Figure 7B & D). The coldest and wettest season (winter) has such a low  $\delta^{18}$ O<sub>w</sub> value (-4.64 ± 0.86 % VSMOW) that it is not represented by the highest  $\delta^{18}O_c$  values, as would be the case if  $\delta^{18}O_c$  would reflect a pure temperature signal (Figure 7B & D). This interpretation of the seasonality in Saiwan is also consistent with the temporal relationship between the clumped isotope sampling locations in specimen HU-027: Sample RB\_1 (winter) comes earliest in the chronology, closely followed by samples R\_8 - R\_11 (spring), and RB\_2 (spring) directly precedes R\_1 -R 7 (summer). The latter two partly overlap later in the chronology (Figure 2). The result is a seasonality in which the temperature cycle and the hydrological cycle (reconstructed through the  $\delta^{18}O_w$  value) are out of phase. The  $\delta^{18}O_c$  value of carbonate precipitated under these conditions therefore exhibits hysteresis behaviour (see Figure 7). Based on the clumped isotope dataset from specimen HU-027 alone, we reconstruct a seasonal sea surface temperature at Saiwan of 19.2 ± 3.8°C to 44.2 ± 4.0°C. The oxygen isotopic composition of the seawater varied from -4.62 ± 0.86 % VSMOW in winter to +0.86 ± 1.6 % VSMOW in summer during the lifetime of specimen HU-027.

## 4.3 Monthly temperature, $\delta^{18}O_w$ and growth rate at Saiwan

612

617

620

624

- While the strong seasonal variability in  $\delta^{18}O_w$  value of the seawater in Saiwan contradicts the typical 636 explanation of  $\delta^{18}O_c$  fluctuations in mollusc shells, the assumptions under which the ShellChron model 637 operates (see **section 2.4.1**) are not violated by this observation. We believe our characterization of 638 seasonal variability in temperature and  $\delta^{18}O_w$  value to be realistic for the following reasons:
- Firstly, this temperature distribution over the year, which is offset in phase from the precipitation seasonality, is observed in modern tropical climates, especially those affected by monsoon-like precipitation seasonality.
- Secondly, regardless of the  $\delta^{18}O_c$ -temperature relationship, our clumped isotope data show that the maximum temperature is still recorded by the minimum  $\delta^{18}O_c$  value. Therefore, the combination of  $\delta^{18}O_c$  measurements and seasonal timing based on ShellChron or Daydacna will still assign the correct temperature,  $\delta^{18}O_w$  and growth rates to  $\delta^{18}O_c$  samples in each part of the year. The nonlinear  $\delta^{18}O_c$ -temperature relationship (**Figure 7**) therefore does not undermine the growth rate-temperature discussion.
- Thirdly, the comparison between results from the Daydacna algorithm (which is independent from the isotope measurements) with our ShellChron results in specimen **B10** shows the same age for this specimen using both independent methods. If ShellChron would under- or overestimate the age of our specimens due to the nonlinear  $\delta^{18}O_c$ -temperature relationship, this would cause a mismatch between these results.
- Finally, a previous study carried out on **B10** (de Winter et al., 2020) presents an analysis of the daily layers in specimen **B10** using multiple lines of evidence. The result of this study is that this specimen records on average 372 daily layers per  $\delta^{18}O_c$  cycle, consistent with astrophysical models of the slow-down of the axial rotation of Earth by friction in the Earth-Moon system, which influences the length of day on geological timescales. The hypothesis that  $\delta^{18}O_c$  cyclicity in *T. sanchezi* represents the full annual cycle (based on ShellChron results) is consistent with this evidence.
  - Considering the above, we combine information from age models with stable and clumped isotope data from specimen **HU-027** to estimate monthly mean temperature,  $\delta^{18}O_w$  and growth rate for all specimens in the dataset based on their  $\delta^{18}O_c$  values (see **section 2.4.4**). This further data analysis step works under the assumption that the clumped isotope data from specimen **HU-027** samples the seasonal variability in temperature and  $\delta^{18}O_w$  in Saiwan. We also assume that the ShellChron age models based on  $\delta^{18}O_c$  profiles in the other specimens are accurate enough to reliably distinguish between the hot, high- $\delta^{18}O_w$  and cooler low- $\delta^{18}O_w$  half of the annual cycle, such that the correct  $\delta^{18}O_w$  value can be estimated for each  $\delta^{18}O_c$  value in the profile and the  $\delta^{18}O_c$  value can be used to estimate paleotemperature at that time of the year. Given the uncertainty of ShellChron age models for the  $\delta^{18}O_c$  profiles in our compilation (~28 days; see **Table 3**), we believe that our age models are accurate enough to do this.
- Figure 8 shows the spread in monthly temperature,  $\delta^{18}O_w$  and growth rate averages. **Table 4** also highlights the implications of the seasonal variability in  $\delta^{18}O_w$  throughout the year in the Saiwan environment (see Figure 7C-D): Correcting  $\delta^{18}O_c$  values for  $\delta^{18}O_w$  variability in all specimens yields an average coldest month mean temperature (CMMT) of  $18.7 \pm 3.8^{\circ}C$  and a warmest month mean temperature (WMMT) of  $42.6 \pm 4.0^{\circ}C$  for the entire dataset. Mean annual average temperatures were  $33.1 \pm 4.6^{\circ}C$ . The uncertainties on these estimates are propagated from the uncertainties of the lowest and highest temperature clusters in the clumped isotope dataset (see **Table 2** and **Figure 7**).

661

665

Using the classic assumption of a seasonally constant  $\delta^{18}O_w$  value of -1 %vSMOW yields a considerably narrower and warmer monthly temperature range (CMMT – WMMT) of 31.8 – 37.6°C. Alternatively, when we exclude the coldest clumped isotope cluster in the **HU-027** dataset, which has a very low  $\delta^{18}O_w$  value of -4.62  $\pm$  0.86 %vSMOW (**Figure 7**), and might therefore be biased by a strong seasonal influx of meteoric water, and assume the mean of the other three clusters (-0.25 %vSMOW) as a constant  $\delta^{18}O_w$  value, the CMMT-WMMT range becomes 35.7 – 41.8°C. Recent studies demonstrated that the assumption of seasonally constant  $\delta^{18}O_w$  values in shallow marine environments often leads to an underestimation of the seasonal temperature range from  $\delta^{18}O_c$  measurements (e.g. de Winter et al., 2021b). We observe the same effect here and therefore use the seasonal temperature reconstructions that take into account seasonal  $\delta^{18}O_w$  variability from our clumped isotope *T. sanchezi* dataset throughout the remainder of the discussion.

**Table 4**: Overview of monthly mean, maximum and minimum estimates of  $\delta^{18}O_c$  (in %VPDB),  $\delta^{13}C$  (in %VPDB), temperature (in °C),  $\delta^{18}O_w$  (in %VSMOW) and growth rate (in μm/day) per specimen. Note that temperature is estimated in three ways to test the sensitivity to different assumptions for the value of  $\delta^{18}O_w$ : Firstly, by using the  $\delta^{18}O_w$ -temperature relationship based on clumped isotope clusters in specimen **HU-027** (see **Fig. 7**). Secondly, by assuming the classical ice-free mean ocean value of -1 %VSMOW. Thirdly, by using the mean  $\delta^{18}O_w$  value based on the heaviest three clumped isotope clusters in specimen **HU-027** (-0.25 %VSMOW; see **Fig. 7B**). In each temperature column, results for the different  $\delta^{18}O_c$  profiles depend on the same assumption of  $\delta^{18}O_w$  variability, so these seasonal temperature estimates are not fully independent from each other. Data given for specimen **HU-027** highlights the spread in temperature

| outcomes betw |           | petwee                 | en               |       | tne                           | 9     |       | cius                                                                           | ters |      | ı                                                          | nstead |      |                                                                          | OT   |      |                                                                         | montnly |       |                       | values. |     |     |
|---------------|-----------|------------------------|------------------|-------|-------------------------------|-------|-------|--------------------------------------------------------------------------------|------|------|------------------------------------------------------------|--------|------|--------------------------------------------------------------------------|------|------|-------------------------------------------------------------------------|---------|-------|-----------------------|---------|-----|-----|
| specimer      | n profile | species                | δ¹*Ο<br>(‰ VPDB) |       | δ <sup>13</sup> C<br>(‰ VPDB) |       |       | Temperature (°C; following δ¹8Oc-T relationship from clumped isotope clusters) |      |      | Temperature<br>(°C; assuming $\delta^{18}O_w = -1$ %vSMOW) |        |      | Temperature (°C; assuming δ <sup>18</sup> O <sub>w</sub> = -0.25 ‰VSMOW) |      |      | δ¹8O <sub>w</sub> (‰VSMOW; reconstructed from clumped isotope clusters) |         |       | Growth rate<br>(μm/d) |         |     |     |
|               |           |                        | mean             | max   | min                           | mean  | max   | min                                                                            | mean | max  | min                                                        | mean   | max  | min                                                                      | mean | max  | min                                                                     | mean    | max   | min                   | mean    | max | min |
| B10           |           | Torreites sanchezi     | -5.37            | -5.04 | -5.73                         | 1.13  | 1.51  | 0.72                                                                           | 31.8 | 43.1 | 20.2                                                       | 35.1   | 37.1 | 33.4                                                                     | 39.2 | 41.2 | 37.4                                                                    | -1.69   | 0.59  | -4.33                 | 28      | 48  | 10  |
| B11           |           | Oscillopha figari      | -3.58            | -3.17 | -3.89                         | 1.70  | 2.01  | 0.99                                                                           | 31.1 | 33.8 | 12.5                                                       | 25.9   | 27.5 | 23.9                                                                     | 29.8 | 31.3 | 27.7                                                                    | 0.02    | 0.82  | -3.89                 | 35      | 59  | 21  |
| В6            |           | Vaccinites vesiculosus | -4.08            | -3.44 | -4.57                         | 0.81  | 1.04  | 0.67                                                                           | 31.6 | 36.2 | 12.2                                                       | 28.4   | 30.9 | 25.2                                                                     | 32.3 | 34.9 | 29.0                                                                    | -0.40   | 0.82  | -4.44                 | 55      | 76  | 37  |
| H576          |           | Torreites sanchezi     | -5.73            | -5.25 | -6.23                         | -0.07 | 0.76  | -1.00                                                                          | 32.8 | 44.2 | 19.9                                                       | 37.1   | 39.8 | 34.5                                                                     | 41.2 | 44.0 | 38.5                                                                    | -1.87   | 0.42  | -4.40                 | 46      | 71  | 23  |
| H579          | Α         | Torreites sanchezi     | -5.92            | -5.43 | -6.38                         | -0.23 | 0.40  | -1.28                                                                          | 37.3 | 45.1 | 22.7                                                       | 38.1   | 40.6 | 35.5                                                                     | 42.2 | 44.8 | 39.5                                                                    | -1.17   | 0.28  | -3.65                 | 43      | 80  | 19  |
| H579          | В         | Torreites sanchezi     | -5.46            | -4.99 | -6.36                         | 0.16  | 0.71  | -0.87                                                                          | 35.0 | 44.9 | 19.9                                                       | 35.6   | 40.5 | 33.1                                                                     | 39.7 | 44.7 | 37.1                                                                    | -1.15   | 0.62  | -4.40                 | 52      | 83  | 0   |
| H579          | С         | Torreites sanchezi     | -5.65            | -4.91 | -6.17                         | 0.26  | 1.42  | -1.19                                                                          | 33.8 | 44.0 | 19.9                                                       | 36.6   | 39.4 | 32.7                                                                     | 40.7 | 43.6 | 36.7                                                                    | -1.59   | 0.53  | -4.40                 | 41      | 116 | 0   |
| H579          | D         | Torreites sanchezi     | -5.49            | -4.81 | -5.90                         | 0.49  | 1.36  | -0.78                                                                          | 29.1 | 43.4 | 19.9                                                       | 35.8   | 38.0 | 32.2                                                                     | 39.9 | 42.1 | 36.2                                                                    | -2.33   | 0.49  | -4.48                 | 53      | 122 | 19  |
| H579          | E         | Torreites sanchezi     | -5.98            | -5.54 | -6.61                         | -0.26 | 0.94  | -1.82                                                                          | 35.9 | 46.3 | 19.3                                                       | 38.4   | 41.8 | 36.0                                                                     | 42.6 | 46.1 | 40.1                                                                    | -1.55   | 0.27  | -4.57                 | 62      | 94  | 4   |
| H579          | average   | Torreites sanchezi     | -5.70            | -5.14 | -6.28                         | 0.08  | 0.97  | -1.19                                                                          | 34.2 | 44.7 | 20.3                                                       | 36.9   | 40.0 | 33.9                                                                     | 41.0 | 44.3 | 37.9                                                                    | -1.56   | 0.44  | -4.30                 | 50      | 99  | 8   |
| H585          |           | Torreites sanchezi     | -5.42            | -4.64 | -6.14                         | -0.21 | 0.50  | -1.23                                                                          | 32.3 | 43.7 | 19.8                                                       | 35.4   | 39.3 | 31.3                                                                     | 39.5 | 43.5 | 35.3                                                                    | -1.65   | 0.82  | -4.44                 | 39      | 60  | 12  |
| HU-027        |           | Torreites sanchezi     | -5.42            | -4.70 | -6.15                         | 1.45  | 2.04  | 0.72                                                                           | 33.9 | 44.2 | 19.2                                                       | 35.4   | 39.3 | 31.6                                                                     | 39.5 | 43.5 | 35.6                                                                    | -1.34   | 0.86  | -4.62                 |         |     |     |
| AVERAGE       |           | -5.28                  | -4.72            | -5.83 | 0.48                          | 1.15  | -0.46 | 33.1                                                                           | 42.6 | 18.7 | 34.7                                                       | 37.6   | 31.8 | 38.8                                                                     | 41.8 | 35.7 | -1.34                                                                   | 0.59    | -4.33 | 45                    | 81      | 15  |     |

Figure 8: Variability in temperature (A-B), estimated  $\delta^{18}O_w$  (C-D) and growth rate (E-F) in all specimens (A, C & E) and in different profiles of specimen H579 (B, D & E). Coloured violin plots indicate the spread in the full dataset, while black symbols indicate monthly mean values. Grey horizontal lines show the highest and lowest monthly mean value, respectively, through the entire dataset (A, C & E) or specimen H579 (B, D & E) with shaded rectangles indicating variability around these values. Note that growth rates could not be estimated for specimen HU-027, and that the black symbols for this specimen in  $\delta^{18}O_w$  and temperature plots (A and C) indicate values for the four clusters (see Table 1) instead of monthly values.

### 4.4 Inter-species differences

There is a significant difference between the  $\delta^{18}O_c$  ranges recorded in *V. vesiculosus*, *O. figari* and *T. sanchezi* in our dataset (see **Figure 8** and **Figure 9**). When considering the seasonal  $\delta^{18}O_w$  variability in Saiwan, the *T. sanchezi* specimens record significantly higher temperatures in their shells (19.2 – 45.1 °C range of monthly temperatures, with an average of 33.6°C) than *V. vesiculosus* (12.2 – 36.2 °C; average: 31.6°C) and *O. figari* (12.5 – 33.8°C; average: 31.1°C). While the age models allow growth stops, it is possible that some months were not recorded by one or more of these species, reducing the seasonal range. This is also evident from the minimum monthly mean growth rate modelled from some  $\delta^{18}O_c$  profiles (e.g. in specimen **H579**; see **Table 4**) being zero.

Nevertheless, the inter-species difference is surprising, given that these specimens originated from the same biostrome (or directly above, in the case of *O. figari*; see **Figure 1B-C**). The lower temperatures (~2

% higher mean  $\delta^{18}O_c$ ) recorded in *O. figari* may be attributed to its slightly higher stratigraphic position. Since the Saiwan site is embedded in a long stratigraphic record and is dated based on ammonite biozones, which yield a dating accuracy on the order of ~100 kyr (Lehmann, 2015), there remains some uncertainty to the (internal) age variability between the units (Kennedy et al., 2000; Schumann, 1995). Therefore, it is possible that the O. figari specimen studied here sampled a different climate and paleoenvironment than the other specimens. This different paleoenvironment may be characterized by a different  $\delta^{18}O_w$  value or seasonal  $\delta^{18}O_w$  range, resulting in a mild underestimation of paleotemperature and its seasonal range, or a true difference in ambient temperature during the lifetime of the oyster compared to the rudists. Since the mean  $\delta^{18}O_c$  value of O. figari is almost 2 % higher than that of T. sanchezi (**Table 4**), mean annual  $\delta^{18}O_w$ would have to have increased by 2 % within a (geologically) very short time interval, given the close stratigraphic relationship between the species, to explain the difference. A plausible way to achieve such a change might be a shift in the seasonal  $\delta^{18}O_w$  regime: If the climate in which *O. figari* grew did not feature the highly  $\delta^{18}O_w$ -depleted winter season (see **Figure 7**) and instead featured year-round  $\delta^{18}O_w$  values close to -0.25 %VSMOW (the mean value of the summer and spring/autumn clusters in HU-027), the difference would explain ~1.2 % of the 2 % difference in the  $\delta^{18}O_c$  value. The remaining 0.8 % offset between  $O_c$ figari and T. sanchezi could then be explained by a drop in mean annual temperature of ~3.5°C over time based on a typical  $\delta^{18}$ O<sub>c</sub>-temperature sensitivity of 4.3-4.5 °C/‰ (Epstein et al., 1953; Grossman and Ku, 1986; Marchitto et al., 2014). In this scenario, the seasonal temperature range experienced by O. figari was roughly 27.7-31.3°C (average temperature of 29.8°C), ~1.3°C lower than the average temperature recorded by O. figari when accounting for the  $\delta^{18}O_w$  seasonality reconstructed by clumped isotope analysis in specimen HU-027, but with a significantly reduced seasonal variability (Table 4). Note that this constant  $\delta^{18}O_w$  scenario would also significantly increase the estimated winter temperatures from O. figari, which are very low (12.5°C; see **Table 4**) when applying the  $\delta^{18}O_w$  seasonality from our clumped isotope data. We therefore consider it likely that the strong drop in temperature and  $\delta^{18}O_w$  value that characterized the winter season in Saiwan may not be recorded in O. figari even if it occurred in the climate and environment of O. figari. The temperature distribution in Figure 9D further confirms this by showing that these exceptionally low winter temperatures in O. figari are represented by a small fraction of the data from this specimen, while the majority of the samples record estimated temperatures between 25°C and 35°C.

715716

724

728

734735

737

739

**Figure 9**: Overview of maximum (**A**) and minimum (**B**) modern monthly SST based on the NOAA SST dataset (Huang et al., 2017) at locations of modern bivalve occurrences (**C**) based on the OBIS dataset (OBIS, 2020) compared with the monthly temperatures reconstructed from the Saiwan fossil molluscs (**D**). The grey rectangle in (**D**) marks the range of temperature limits for modern bivalves in the hottest shallow marine environment studied in (Compton et al., 2007): Roebuck Bay.

This stratigraphic argument cannot explain the temperature differences between V. vesiculosus and T. sanchezi, since they were sampled in life position from the same biostrome. Given the rapid growth of these organisms (de Winter et al., 2020; de Winter et al., 2017) and the rapid build-up and succession of the biostromes they are found in (Gili et al., 1995; Gili and Skelton, 2000; Ross and Skelton, 1993), it seems likely that these organisms lived within geologically short time periods from each other (conservatively less than 1000 years), and therefore sampled a similar climate in the Saiwan region. This seems plausible considering the comparatively large overlap in the frequency distribution of temperatures recorded by both species (Table 4; Figure 9D). The growth rate of V. vesiculosus was on average similar to that of T. sanchezi (see Table 4 and Figure 8E) and the spatial sampling resolution in V. vesiculosus (250 μm) was often higher than that in T. sanchezi shells (~100-500 μm (de Winter et al., 2017). Thus, it seems unlikely that the differences between the seasonal ranges obtained from the shells of both species can be attributed to undersampling of the full recorded temperature of V. vesiculosus, a source of variability discussed in (de Winter et al., 2021a; Goodwin et al., 2003; Judd et al., 2017). Applying the high seasonal range in  $\delta^{18}O_w$  observed in **HU-027**, the maximum monthly temperature recorded by *V. vesiculosus* (36.2°C) is at least 5 degrees lower than the maximum monthly temperatures recorded in the T. sanchezi specimens (>43°C in all specimens, average of 44°C; **Table 4**). At the same time, mean annual temperatures

762

in *V. vesiculosus* (31.6°C) and *T. sanchezi* (33.0°C) differ by less than 1.5 degrees. As in *O. figari*, winter temperatures in *V. vesiculosus* are exceptionally low compared to temperature ranges in other specimens in the assemblage (**Table 4**) and the temperature distribution in this specimen (**Figure 9D**). This is a consequence of the fact that winter datapoints in the *V. vesiculosus* dataset are associated with the very low  $\delta^{18}O_w$  values reconstructed from the winter datapoints in the clumped isotope dataset, resulting in low temperature estimates when applied on the  $\delta^{18}O_c$  values measured in *V. vesiculosus*. Since all *T. sanchezi* specimens yield consistent winter temperatures of ~20°C, we consider it plausible that *V. vesiculosus* did not record the seasonally low  $\delta^{18}O_w$  conditions and conclude that winter SSTs in Saiwan are more accurately estimated by the *T. sanchezi* data (19-20°C).

The growth rate of *T. sanchezi* specimens is diminished when the range of maximum tolerable temperatures (upper thermal limits) for modern bivalves occurring in hot, shallow marine settings (Roebuck Bay, Australia, tolerable thermal range: 32.7 – 41.8°C; (Compton et al., 2007)) is reached or exceeded (see **Supplement S6** in the Zenodo repository). While the frequency of temperatures recorded by *V. vesiculosus* and *O. figari* specimens in our dataset decreases when these maximum tolerable temperatures are reached (**Figure 9**), temperatures recorded in *T. sanchezi* specimens regularly exceed modern temperature maxima. Higher temperatures recorded in *T. sanchezi* specimens suggest that the upper thermal tolerance of this species may exceed those recorded in modern bivalves and that *T. sanchezi* had a higher temperature tolerance than *V. vesiculosus* and *O. figari*.

This conclusion is further supported by the observation that almost all studied *T. sanchezi* specimens record a monthly temperature range that exceeds that of *V. vesiculosus* (and *O. figari*; **Table 4**) and that the monthly temperature range recorded by *T. sanchezi* specimens is consistent (see **Figure 8B** and **Table 4**). The fact that all *T. sanchezi* specimens together record over 30 years of seasonality with consistently low summer  $\delta^{18}O_c$  values (monthly minima of -5.7 %VPDB or lower, corresponding to maximum monthly temperatures >40°C; see **Figure 3 & Table 4**) also shows that these conditions were not isolated events but a persistent feature of the local climate in Saiwan during the Campanian. Finally, daily rhythms in the shell chemistry of the large *T. sanchezi* specimen **B11** have previously revealed that this specimen likely grew year-round, and thus recorded the full seasonal temperature cycle (de Winter et al., 2020).

An important limitation to this inter-species paleoclimate comparison is that, while the  $\delta^{18}O_c$  profiles presented here record independent records of seasonal variability in different specimens, the interpretation of this  $\delta^{18}O_c$  variability in terms of paleotemperature relies on clumped isotope data collected in one specimen (HU-027). Our assumption that all rudist specimens occurred in the same biostrome and sampled a highly similar climate allows us to use the seasonal  $\delta^{18}O_w$  structure inferred from the clumped isotope dataset in HU-027 to obtain temperature seasonality from  $\delta^{18}O_c$  profiles. Therefore, the temperature reconstructions from different  $\delta^{18}O_c$  profiles are not fully independent.

the temperature reconstructions from uniferent of Giptomes are not runy independent.

4.5 Campanian temperature extremes compared to modern climates – Implications for temperature
 tolerance in past shallow marine environments

**Figure 9A-C** presents an overview of the data on the occurrence of modern bivalves from the OBIS dataset (OBIS, 2020) cross-referenced with the monthly temperatures in these modern locations based on NOAA SST data (Huang et al., 2017) for the years 1981-2010. The lowest monthly temperature in environments containing modern bivalve recordings is -1.8°C. The maximum SST of the warmest month recorded in the NOAA dataset at any location in which modern bivalves are reported is 34.1°C. Based on the combination of these datasets, it becomes clear that the Saiwan environment was warmer than any shallow marine

environment that sustains bivalves in the modern world (**Figure 9D**). The range of temperatures experienced by bivalves in the warmest setting studied by (Compton et al., 2007; 32.7 - 41.8 °C) is frequently exceeded in the Saiwan environment, according to the over 4°C higher mean WMMT estimated from our data ( $42.6 \pm 4.0$  °C; **Table 4**; **Figure 8A**). The warmest monthly temperatures in the Saiwan environment thus commonly exceed the maximum monthly temperatures in the living environments of modern bivalves, even when we consider the mean warmest monthly temperature for all specimens instead of the monthly extremes recorded by some *T. sanchezi* specimens in our dataset (44°C). When we consider that bivalves are known to stop growing their shell during stressful periods in the year (e.g. winter in most modern bivalves; Ivany (2012) and summer in some hotter climates Buick and Ivany (2004)), it may be possible (though perhaps unlikely) that the seasonal ranges recorded in our specimens from Saiwan underestimate the true seasonal temperature range in this paleo-environment.

Our clumped isotope data corroborates evidence based on stable oxygen isotope profiles through lowlatitude Tethyan rudists (with assumptions of constant seawater  $\delta^{18}O_w$  value; e.g. (de Winter et al., 2017; Steuber et al., 2005) that shallow seas in the Cretaceous Tethys Ocean margins warmed up to temperatures unseen in modern climates (Figure 9), but are not unheard of in the fossil record (e.g. Paleocene-Eocene Thermal Maximum tropical mean annual SSTs >40°C; (Aze et al., 2014)). Clumped isotope data from T. sanchezi specimen HU-027 also reveals that assumptions of year-round constant  $\delta^{18}$ O<sub>w</sub> value can be very far off from the true  $\delta^{18}$ O<sub>w</sub> seasonality: In the Saiwan site, the coldest season recorded in **HU-027** is characterized by a mean  $\delta^{18}O_w$  value of -4.62  $\pm$  0.86 %VSMOW, which is more than 5‰ lighter than the mean  $\delta^{18}$ O<sub>w</sub> value during the warmest season (0.86 ± 1.61 ‰VSMOW). If  $\delta^{18}$ O<sub>w</sub> would be considered constant year-round (e.g. -1 %VSMOW),  $\delta^{18}O_c$ -based temperature reconstructions would underestimate the true seasonality at this site during the Campanian by over 10°C (see discussion in section 4.3 and results in Table 4). Our data instead shows that the low-latitude Saiwan paleoenvironment experienced a higher seasonal temperature range than the roughly contemporary (78 Ma) higher midlatitudes of the Campanian boreal chalk sea recorded in shells from the Kristianstad basin (18.7 ± 3.8 - $42.6 \pm 4.0$  °C, or  $23.9 \pm 6.4$  °C seasonal range for Saiwan vs  $15.3 \pm 4.8 - 26.6 \pm 5.4$  °C, or  $11.2 \pm 7.3$  °C seasonal range for Kristianstad basin), for which seasonal clumped isotope reconstructions were previously presented (de Winter et al., 2021b).

Note that the absolute mean annual temperature for the Saiwan ecosystem (33.1 ± 4.6°C) was significantly higher than that of the higher latitude Boreal Chalk sea (20.1 ± 1.3°C; de Winter et al. (2021b)), yielding a temperature gradient of ~13°C over the latitudes 3°S-50°N. The present-day mean annual sea surface temperatures at the closest weather stations to these localities are 8.2°C (Kristianstad, Sweden, 56°N (de Winter et al., 2021b; Huang et al., 2017)) and 27.9°C (Muscat, Oman, 23°N; (World Sea Temperatures, 2024)), yielding a present-day SST gradient of ~19°C, more than 1.5x that for the same locations in the Campanian. This result corroborates previous evidence from data and models that the latitudinal temperature gradient was smaller compared to the present-day during periods of warmer climate, such as the Campanian (Amiot et al., 2004; Burgener et al., 2018). However, it must be noticed that the paleolatitudes of these sites are slightly different than their modern latitudes.

The Campanian climate in Saiwan is significantly different from that of present-day Oman, which experiences a sea surface temperature seasonality of  $24.2 \pm 1.6 - 31.5 \pm 1.6$ °C (World Sea Temperatures, 2024) and a very limited seasonal precipitation range (0 – 11 mm/month; (climate-data.org, 2024)). The seasonal pattern in temperature and seawater composition in the Campanian is likely caused by large seasonal variability in precipitation, which is common in present-day tropical climates. In this climate,

seasons with hot and relatively dry conditions (summers) are interchanged with cooler seasons featuring an influx of isotopically light water, which diluted the shallow seawater at Saiwan or produced a layer of lower salinity waters close to the sea surface, which was recorded by these very shallow-dwelling photosymbiotic rudists.

The above-mentioned conditions make summers in the Saiwan paleoenvironment hot, even for the Campanian with its global mean annual temperature of 20-25°C (O'Brien et al., 2017). *T. sanchezi* apparently thrived under these conditions while *V. vesiculosus* and *O. figari* stopped producing their shell at temperatures close to those that limit modern shallow marine bivalves (~34°C; Clarke (2014); Compton et al. (2007)) and perhaps also during the high-precipitation and lower-salinity phases of the winter season. This suggests that *T. sanchezi* may have been particularly well-adapted to the high seasonality in temperature and precipitation and hot summers in its environment. This hypothesis is further supported by the observation that the genus *Torreites* occurs exclusively in the late Cretaceous low latitudes (18°N – 29°N) of the near East and middle America (Global Biodiversity Information Facility, 2024). The unexpectedly high seasonal temperature variability in the Saiwan paleoenvironment (18.7  $\pm$  3.8 – 42.6  $\pm$  4.0 °C; **Table 4**; **Figure 8**) might have provided relief for the Saiwan molluscs, since they were not forced to complete their entire life cycle at exceptionally high temperatures and may have recovered from heat exposure during the cooler seasons.

While the conditions in Campanian Saiwan approach the limits of what has been observed for present-day eukaryotes (Clarke, 2014), recent growth experiments on modern gastropods show that temperatures up to 45°C can be tolerated by molluscs for limited amounts of time (Prayudi et al., 2024). Organisms that need to withstand these stressful heat conditions often develop specialized proteins and enzymes to continue bodily functions while experiencing close to lethal temperatures (Tehei et al., 2005; Tehei and Zaccai, 2007). While the Campanian fossils studied here do not preserve remnants of these organic molecules, these molluscs probably employed similar strategies to survive through the extreme summer heat. We thus hypothesize that the members of the Saiwan ecosystem (especially *T. sanchezi*) must have been evolutionarily adapted to its seasonally hot environment. These observations raise the question whether, given enough time, multicellular organisms such as shallow marine bivalves may survive life in hotter climates, and to which degree the thermal tolerance of modern relatives can be used as a reference point for interpreting the fossil record. More research into extreme paleo-communities such as the Saiwan ecosystem is crucial for understanding the evolutionary limits to which metazoans can adapt to warmer climates, how much time is needed to make these adaptations, and whether such adaptation can come in time to save modern shallow marine communities from rapid climate change.

### **Acknowledgements**

The authors would like to acknowledge the assistance of Leonard Bik with sample preparation. Arnold van Dijk and Desmond Eefting assisted with the stable isotope analyses in the Utrecht University lab. Stijn van Malderen is acknowledged for his assistance with the LA-ICP-MS measurements. This study has benefitted from discussions with Barbora Krizova, Barbara Goudsmit-Harzevoort, Jingjing Guo and Tobas Agterhuis during data processing of the clumped isotope results and from discussions with Pim Kaskes about the calibration and interpretation of micro-XRF results.

PC acknowledges funding from the Flemish Research Council (FWO; grant nr. G038022N). PC and SG also thank FWO for the financial support for the VUB XRF platform, as well as the VUB Strategic Program for lab maintenance. NF was supported by a DAAD fellowship during her stay in the Netherlands and received funding from the EU Erasmus program to fund her lab visit. NJW is supported by an NWO VENI grant (grant nr.: VI.Veni.222.354) and chemical analyses carried out during this project were financially supported by an FWO junior postdoc grant (grant nr. 12ZB220N) and MSCA Individual Fellowship (UNBIAS; 843011) both awarded to NJW. *Torreites sanchezi* specimen HU-27 (BSc-thesis N. al-Fudhaili) was examined with micro-Raman spectroscopy at the University of Padova and we would like to acknowledge the help of Claudio Mazzoli, further Bryan Shirley at University of Erlangen (FAU-GZN) helped with EDS-mapping. HU-27 stable isotopes were kindly measured at FAU-GZN by Daniele Lutz and Michael Joachimski.

# Data availability

All chemical data and annotated Python (for the Daydacna routine) and R (for the remaining workflow) scripts used to carry out the data processing for this study are available in Supplement S7 to this publication as well as through the open-access repository (https://doi.org/10.5281/zenodo.12567712). Information about access to specimens H576, H579 and H585 is provided in (Steuber, 1999). Specimens B10 and B11 are archived in the Natural History Museum of Maastricht (the Netherlands), and specimen B6 is archived in the Oertijdmuseum in Boxtel (the Netherlands). Specimen HU-027 is archived at Geozentrum NordBayern at the Friedrich-Alexander Universität in Erlangen (Germany) and the material can be accessed by contacting Matthias López Correa (matthias.lopez@fau.de) or Axel Munnecke (axel.munnecke@fau.de).

### **Author contribution**

NJW, NF and MLC designed the study. NF carried out laboratory measurements under supervision of NJW, MLC, AM and MZ. IA, SG, PC, MLC, JS and MZ provided access to laboratories and assistance with methods. PC, AM and MZ secured funding to support laboratory measurements. RF, JJ, MLC, JS and NJW provided access to sample material and sedimentological context. NJW and NF drafted the first version of the manuscript. All authors were involved in writing and reviewing of subsequent manuscript versions.

### **Competing interests**

The authors declare that they have no conflict of interest.

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
