# Peer review of "Living on the edge: Response of Late Cretaceous rudist bivalves (Hippuritida) to hot and highly seasonal"

_EGUsphere, 2025_

## Referee Comment (RC2)

**Living on the edge: Response of rudist bivalves (Hippuritida) to hot and highly seasonal climate in the**
**low-latitude Saiwan site, Oman**

Niels J. de Winter[1,2], Najat al Fudhaili[3], Iris Arndt[4], Philippe Claeys[2], René Fraaije[5], Steven Goderis[2], John
Jagt[6], Matthias López Correa[7], Axel Munnecke[7], Jarosław Stolarski[8], Martin Ziegler[9]

[revised manuscript text omitted]
 through *T. sanchezi* specimens (**A, B, C, D** & **F**; green frame) and one profile of both $\delta^{18}O_c$ and $\delta^{13}C$ through *V. vesiculosus* specimen **B6** (**E**; blue frame) and *O. figari* specimen **B11** (**G**; red frame). Vertical axes of *T. sanchezi* profiles in **A-D** are equal, while *V. vesiculosus*, *O. figari* and *T. sanchezi* specimen **HU-027** have different vertical axes. Records in **D** represent parallel profiles through the same specimen (**H579**). The shaded background colours represent time of year based on the ShellChron chronologies constructed using these $\delta^{18}O_c$ and $\delta^{13}C$ profiles, with darker colour indicating samples assigned to days earlier in the year. Black dots in **G** show $\delta^{18}O_c$ and $\delta^{13}C$ values associated with clumped isotope measurements in **HU-027** and colored dots and error bars indicate the spread in $\delta^{18}O_c$ and $\delta^{13}C$ and location of the material used in clumped clusters presented in **Figure 7**.

3.2 Trace element results

[Figure]

Trace element analyses highlight that shells of all three species are generally characterized by low concentrations of Mn and Fe (**Figure 5**). *T. sanchezi* exhibits median Mn concentrations of 38 µg/g (average: 50 ± 44 µg/g, 1σ) and median Fe concentrations of 56 µg/g (average: 77 ± 116 µg/g, 1σ) with a few isolated locations in the shell with concentrations exceeding 300 µg/g and 1000 µg/g for Mn and Fe, respectively. The carbonate in *V. vesiculosus* has somewhat higher median Mn concentrations of 176 µg/g (average: 192 ± 94 µg/g, 1σ) and median Fe concentrations of 125 µg/g (average: 173 ± 152 µg/g, 1σ) with some locations showing Mn and Fe concentrations exceeding 500 µg/g and 800 µg/g, respectively. A clear positive trend is observed towards higher Mn and Fe concentrations in *V. vesiculosus* samples (**Fig. 5A**). Finally, *O. figari* has median Mn concentrations of 63 µg/g (average: 75 ± 33 µg/g, 1σ) and much lower median Fe concentrations of 4 µg/g (average: 5.7 ± 5.8 µg/g, 1σ), with maximum Mn and Fe concentrations of 180 µg/g and 40 µ
[revised manuscript text omitted]

---

## Author Response (AR1)

**Author comment(s) Preprint egusphere-2025-2308**

**AC1**

Dear Antje Voelkner, dear reviewers,

We appreciate the supportive and helpful comments by both reviewers on our manuscript. The feedback by both reviewers is minor and can be addressed quite easily in our opinion. Below, we detail how we address the points raised by Reviewer 1 in point-by-point fashion. Line numbers refer to the originally submitted version of the manuscript (as in the reviewer's report). In addition, we are happy to submit an annotated version of our manuscript on resubmission in which we track all changes made in reply to these comments. We trust that the suggested changes will address the concerns raised by the reviewer on the current manuscript version and make our contribution suitable for publication in Climate of the Past.

**Reviewer 1:**

The manuscript would be an excellent addition to this journal upon some moderate revisions. The manuscript exhibits excellent scientific significance, as it explores seasonal climate extremes during Cretaceous greenhouse climates. Relating this information to modern bivalves and their maximum thermal tolerance provides excellent context for why this study is important and the requirement for additional research. The quality of the work is sound. The implication of the results is appropriate and within reason, given the data. The presentation quality is good, but could be improved. Figures and figure captions need to be revised to be clearer. Sections of the manuscript require reorganization to enhance clarity. More detailed revisions are provided below. It is in my opinion that this manuscript is accepted subject to minor revisions.

We thank the reviewer for their positive reception of our manuscript and constructive comments on the presentation of our data and figures. We hope that our suggested changes make our manuscript more accessible to the reader.

**Moderate Comments:**

Figures 2, 3, 4, 5, 8, and 9 all have red and green color elements that need to be changed.

We appreciate that our current color scheme may not be accessible for colorblind readers and thank the reviewer for bringing that to our attention. We propose different solutions to this issue for different figures to preserve a consistent color scheme throughout the manuscript. To choose an accessible and inclusive color scheme, we used the Colorbrewer tool (<a href="https://colorbrewer2.org/">https://colorbrewer2.org/</a>), and we cite the unique HEX codes of the colors we plan to use in our responses below.

- Figure 2: All red text, lines and boxes in panels A and B will be changed to grey. Green and red colors used to indicate calcite and silica mineralogies will be changed to light green (HEX: #66c2a5) and orange (HEX: #fc8d62), respectively (Using "Set2" in the Colorbrewer tool).

- Figures 3, 4, 5, 8 and 9: The green, blue and red color scheme used for the three species (*T. sanchezi*, *V. vesiculosus* and *O. figari*, respectively), will be changed according to scheme "Dark2" of the Colorbrewer tool, using green (HEX: #1b9e77), purple (HEX: #7570b3) and orange (HEX: #d95f02) for these three species, respectively. We will retain the red and blue colors marking carbon and oxygen isotope values in Figure 4 and 6, and the blue-yellow-red diverging color scheme ("RdYlBu" in the Colorbrewer tool) used for clumped isotope temperatures in Figure 4 and 7, as well as the green shading in Figure 8B, D and F ("Greens" color scheme in Colorbrewer tool).

Figure 4. The caption says 9 profiles, but I count 7 unique samples with 11 time series.

We realize that this caption is slightly confusing, but there are indeed 9 profiles through *T. sanchezi* shells and one profile to both *V. vesiculosus* and *O. figari*. To clarify that this is what we mean, we will rephrase the first sentence of the caption as follows:

"Figure 4: Overview of 11 incrementally sampled stable oxygen ( $\delta^{18}$ Oc; blue) and carbon isotope ( $\delta^{13}$ C; red) profiles: 9 profiles through 5 *T. sanchezi* specimens, of which 5 parallel profiles through specimen H579 (A, B, C, D & F; green frame), one profile through *V. vesiculosus* specimen B6 (E; purple frame) and one profile through *O. figari* specimen B11 (G; orange frame)."

Figures 4, 5, and 8. The X- and Y-axis labels need to be made larger for improved legibility.

We will increase the font size of the axes of these figures to improve legibility

Figure 8.  $\delta^{18}$ O is listed as d18O in the x-axis of plots C and D. This should be changed to the correct format.

The vertical axis label in panels C and D of this figure will be updated to reference the correct delta symbol notation.

Font size for Table 4 needs to be increased.

We will attempt to increase the font size of the text in this table as much as possible while still allowing the table to fit on the page. If the font size needs to be increased further, we provide the raw data of the table and suggest to resolve this issue with the type editor.

Line 775. Figure 10 is mentioned but not found in the manuscript.

Thank you for pointing this out. The figure reference was left over from a previous manuscript version and should instead refer to Figure 9. This will be corrected.

Minor Comments

Lines 74-78. Break into two sentences.

This sentence will be divided into two sentences as follows:

"For example, some of these past environments, most notably shallow marine ecosystems, are thought to have reached temperatures exceeding the temperature range of modern equivalent ecosystems (de Winter et al., 2020; Huang et al., 2017; Jones et al., 2022). These temperatures probably exceeded the maximum temperature tolerance at which modern shallow marine species can complete their life cycle, which is typically estimated in the order of 38-42°C (Compton et al., 2007)."

Lines 573-577. Break into two sentences.

This sentence will be divided into two sentences as follows:

"Instead, a positive correlation between  $\delta^{18}O_c$  and  $\delta^{13}C$  values is often observed in modern (non-diagenetically altered) photosymbiotic species, such as tridacnids (Elliot et al., 2009; Killam et al., 2020). Such a correlation has been proposed to be caused by seasonal changes in the isotopic composition of the dissolved inorganic carbon pool due to variability in the activity of photosymbionts in phase with the seasonal effect of temperature on the oxygen isotope composition of the shell (Elliot et al., 2009; McConnaughey and Gillikin, 2008)."

Line 670. Change classical to classic

This will be changed accordingly

Lines 851-856. Break into two sentences.

This sentence will be divided into two sentences as follows:

"T. sanchezi apparently thrived under these conditions while V. vesiculosus and O. figari stopped producing their shell at temperatures close to those that limit modern shallow marine bivalves (~34°C; Clarke (2014); Compton et al. (2007)) and perhaps also during the high-precipitation and lower-salinity phases of the winter season. This suggests that T. sanchezi may have been particularly well-adapted to the high seasonality in temperature and precipitation and hot summers in its environment."

**AC2**

Dear Antje Voelkner, dear reviewers,

We appreciate the supportive and helpful comments by both reviewers on our manuscript. The feedback by both reviewers is minor and can be addressed quite easily in our opinion. Below, we detail how we address the points raised by Reviewer 2 in his annotated PDF in point-by-point fashion. Line numbers refer to the originally submitted version of the manuscript (as in the reviewer's report). In addition, we are happy to submit an annotated version of our manuscript on resubmission in which we track all changes made in reply to these comments. We trust that the suggested changes will address the concerns raised by the reviewer on the current manuscript version and make our contribution suitable for publication in Climate of the Past.

**Reviewer 2 (Werner Piller):**

The manuscript presents an excellent study on rudist and ostreid bivalves from the late Campanian. Since the data are dealing with a warmhouse climate the results and conclusions are of wide relevance. The methods applied are adequate and state-of-the-art. The explanation of the methods is excellent and can also be followed by non-specialists. The interpretation of the results are sound and of great importance since they not only provide information on temperature but also on (paleo)biological aspects in respect to the high temperatures which cause shut off of growth of some taxa and continuous growth in others. The paper is therefore not only a geochemical approach but combines geochemical results with critical (paleo)biological parameters.

We appreciate the thoughtful endorsement of our study by the reviewer and their constructive comments on our text in the annotated PDF, which we plan implement to improve our manuscript.

I made several comments directly in the pdf file which is attached. I suggest to carry out minor changes before the manuscript should be accepted.

We directly implemented all comments pertaining to grammar, spelling and interpunction and briefly reply below to the written comments by the reviewer:

Line 127: We will mention specifically in the caption of Figure 1 that the marly layer referred to here is the layer in between the two biostromes indicated in green in Figure 1B:

"The marly layer containing *O. figari* referred to in the text is the brown layer in between the two green members in the red box."

Line 139: We will rephrase this to "longitudinal cross section" for clarity. This was originally implied with the statement "through the axis of maximum growth" later in the sentence, but we appreciate that just "cross" section may cause confusion. Throughout the revised manuscript, "cross section" will be written without the hyphen.

Line 525: The first use of "sample" will be removed here to prevent repetition of the word in this sentence and promote clarity.

Line 549: We appreciate the suggestion of citing a textbook here for the diagenetic alteration discussion, but we retain the current citations since these are landmark studies of calcite shell

| relevant to this study's specimens. |  |
|-------------------------------------|--|
|                                     |  |
|                                     |  |
|                                     |  |
|                                     |  |
|                                     |  |
|                                     |  |
|                                     |  |
|                                     |  |
|                                     |  |
|                                     |  |
|                                     |  |
|                                     |  |
|                                     |  |
|                                     |  |
|                                     |  |
|                                     |  |
|                                     |  |
|                                     |  |
|                                     |  |
|                                     |  |
|                                     |  |
|                                     |  |
|                                     |  |
|                                     |  |
|                                     |  |
|                                     |  |
|                                     |  |
|                                     |  |
|                                     |  |
|                                     |  |
|                                     |  |

preservation and deal specifically with rudist shells, making them, in our opinion, more directly